# ADM-v2: Pursuing Full-Horizon Roll-out in Dynamics Models for Offline Policy Learning and Evaluation

**Haoxin Lin**[1], **Siyuan Xiao**[1], **Yi-Chen Li**[1], **Zhilong Zhang**[1],
**Yihao Sun**[2], **Chengxing Jia**[1], **Yang Yu**[1]*

[1]National Key Laboratory for Novel Software Technology, Nanjing University, Nanjing, China
& School of Artificial Intelligence, Nanjing University, Nanjing, China
[2]University of Montréal, Montréal, Canada

{linhx,xiaosy,liyc,zhangzl}@lamda.nju.edu.cn
yihao.sun@mila.quebec, {jiacx,yuy}@lamda.nju.edu.cn

## Abstract

Model-based methods for offline Reinforcement Learning transfer extensive policy exploration and evaluation to data-driven dynamics models, effectively saving real-world samples in the offline setting. We expect the dynamics model to allow the policy to roll out full-horizon episodes, which is crucial for ensuring sufficient exploration and reliable evaluation. However, many previous dynamics models exhibit limited capability in long-horizon prediction. This work follows the paradigm of the Any-step Dynamics Model (ADM) that improves future predictions by reducing bootstrapping prediction to direct prediction. We structurally decouple each recurrent forward of the RNN cell from the backtracked state and propose the second version of ADM (ADM-v2), making the direct prediction more flexible. ADM-v2 not only enhances the accuracy of direct predictions for making full-horizon roll-outs but also supports parallel estimation of the any-step prediction uncertainty to improve efficiency. The results on DOPE validate the reliability of ADM-v2 for policy evaluation. Moreover, via full-horizon roll-out, ADM-v2 for policy optimization enables substantial advancements, whereas other dynamics models degrade due to long-horizon error accumulation. We are the first to achieve SOTA under the full-horizon roll-out setting on both D4RL and NeoRL. The code is available at https://github.com/LAMDA-RL/adm2.

## 1 Introduction

Offline Reinforcement Learning (Fujimoto & Gu, 2021; Kumar et al., 2020) is a promising way to solve real-world decision-making tasks. The core challenge lies in learning a well-performing policy from a fixed dataset and conducting off-policy evaluation in the offline training process. Model-based Reinforcement Learning (MBRL) (Yu et al., 2020; 2021; Sun et al., 2023; Lin et al., 2025) in the offline setting is well-suited to addressing the challenge of limited data, since many policy exploration and evaluation can happen within dynamics models. The ideal case is that the dynamics model can roll out complete episodes, acting as a simulator that closely resembles the real world.

An important role of full-horizon roll-outs is to enable the policy to sufficiently explore, which is beneficial for achieving better performance. Sims et al. (2024) finds that the truncation of roll-outs hinders the policy from reaching some edge states, thereby impacting the estimation of the value function. Exploration of edge states facilitates more thorough policy learning and yields further improvements. Additionally, Lu et al. (2022) finds that longer roll-outs combined with a suitable uncertainty penalty lead to better performance. On the other hand, offline policy evaluation (Fu et al., 2021; Yang et al., 2020; Chen et al., 2024b) also imposes requirements on the model's ability to make long-horizon roll-outs. With a data-driven dynamics model, the most natural approach for policy evaluation is to sample several complete trajectories in the model and estimate the expected return.

---

*Corresponding Author

Although many prevalent MBRL algorithms (Yu et al., 2020; Sun et al., 2023) are theoretically derived under the assumption of full-horizon model roll-outs, practically they adopt short and branched roll-outs (Janner et al., 2019). This is because their dynamics models rely on the bootstrapping prediction, which attributes the next state to the prediction of the current state and causes model error accumulation (Xu et al., 2021) during roll-outs. This limitation is commonly faced in almost all popular dynamics models, including ensemble models (Chua et al., 2018; Yu et al., 2020), causal models (Zhu et al., 2022), and adversarial models (Chen et al., 2023; Bhardwaj et al., 2023). For them, roll-out truncation is a compromise to ensure the reliability of model-generated samples.

In this paper, we seek to extend the roll-out length to match the episode horizon and follow the Any-step Dynamics Model (ADM) (Lin et al., 2025). ADM directly models the state transitions resulting from executing any-step actions through a GRU cell (Cho et al., 2014), showing the effectiveness of improving future predictions by reducing bootstrapping prediction to direct prediction. ADM generalizes the direct state transition after executing a multi-step action sequence (Asadi et al., 2018; 2019; Che et al., 2018; Machado et al., 2023) into an any-step formulation. Specifically, ADM takes as input a starting state $s_0$ and an action sequence of arbitrary length $k$, $(a_0, a_1, \cdots, a_{k-1})$, to predict $s_k$. To match the length of the action sequence, ADM duplicates the starting state multiple times. Therefore, the actual input to the GRU inside ADM is a state-action sequence of the form $([s_0, a_0], [s_0, a_1], \cdots, [s_0, a_{k-1}])$, making the network architecture appear inflexible.

The structure of the original ADM not only induces strong coupling between the GRU's hidden representations and the starting state but also prevents any-step prediction from being accelerated through parallel computation. To address these issues, we propose the second version of ADM (ADM-v2). ADM-v2 encodes the starting state $s_0$ into a latent vector $h_0$, which is directly used as the initial hidden state of the GRU. The information of $s_0$ is contained within $h_0$, so each forward of the GRU no longer takes $s_0$ as input but only the action. ADM-v2's decoupling of $s_0$ reduces the impact of perturbations in $s_0$ on multi-step direct prediction, thereby improving the prediction reliability. Moreover, ADM-v2 discards the backtracking mechanism (Lin et al., 2025) proposed in ADM to obtain any-step prediction. We design a roll-out algorithm for ADM-v2 to enable parallel computation of direct predictions from different strides, called **P**arallel **A**ny-step **Roll**-out (PARoll). PARoll also allows the parallel estimation for any-step uncertainty (Lin et al., 2025), which is efficient for policy learning and evaluation with full-horizon roll-outs in offline settings.

In general, our contributions are summarized as follows. (1) We propose the ADM-v2 and the roll-out algorithm PARoll. The experiments on D4RL (Fu et al., 2020) dataset demonstrate the advantages of ADM-v2 in terms of prediction accuracy and efficiency, especially during full-horizon roll-outs. (2) We evaluate the given policy offline by rolling out complete episodes in ADM-v2. Results on DOPE (Fu et al., 2021) benchmark show that this approach yields more accurate policy evaluation than other off-policy evaluation methods. (3) We use the full-horizon roll-outs sampled in ADM-v2 for policy learning and apply any-step uncertainty as a penalty term in Q-value estimation. This algorithm, ADM2PO-fh, achieves SOTA performance on both the D4RL and NEORL (Qin et al., 2022) datasets.

## 2 PRELIMINARIES

### 2.1 MARKOV DECISION PROCESS

A Markov Decision Process (MDP) is defined by the tuple $\mathcal{M} = (\mathcal{S}, \mathcal{A}, T, \rho_0, \gamma)$. Here, $\mathcal{S}$ is the state space, $\mathcal{A}$ is the action space, and $T(s_{t+1}, r_{t+1}|s_t, a_t)$ denotes the probability density function of the next state $s_{t+1}$ and the reward $r_{t+1}$ given state $s_t$ and action $a_t$. $\rho_0$ is the initial state distribution over $\mathcal{S}$. $\gamma \in [0, 1)$ is the discount factor. We use $\rho^\pi$ to denote the on-policy distribution over states induced by the dynamics function $T$ and the policy $\pi$. The objective of RL is to learn an optimal policy $\pi^\star$ that maximizes the expected discounted return $\mathbb{E}_{\rho^\pi} \left[ \sum_{t=1}^{\infty} \gamma^{t-1} r_t \right]$.

### 2.2 OFFLINE MBRL

In the offline setting, the agent only has access to a fixed dataset of transitions $\mathcal{D}_{\text{env}} = \{(s_t, a_t, r_{t+1}, s_{t+1})\}$, collected by the behavior policy $\pi_b$ in the real world. In the offline MBRL framework, a dynamics model $\hat{T}$ is learned from $\mathcal{D}_{\text{env}}$ by maximizing the expected likelihood $\mathbb{E}_{(s_t, a_t, r_{t+1}, s_{t+1}) \sim \mathcal{D}_{\text{env}}}[\log \hat{T}(s_{t+1}, r_{t+1}|s_t, a_t)]$. This allows us to construct a surrogate MDP

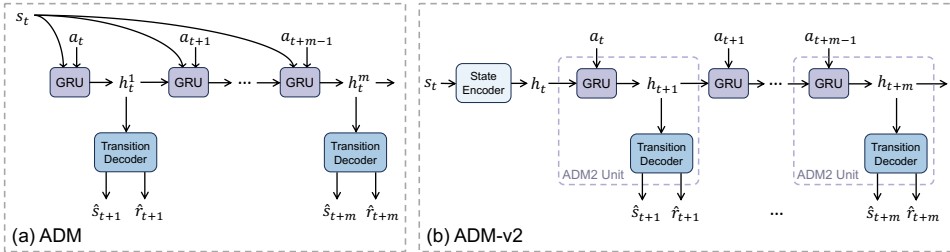

Figure 1: Illustration of (a) the vanilla Any-step Dynamics Model (Lin et al., 2025); (b) the second version of Any-step Dynamics Model (ADM-v2), both are structured using a GRU Cell.

$\hat{M} = (\mathcal{S}, \mathcal{A}, \hat{T}, \rho_0, \gamma)$. Both the offline dataset $\mathcal{D}_{\text{env}}$ and the rolled-out data by the learned model $\hat{T}$, denoted as $\mathcal{D}_{\text{model}}$, are utilized to optimize the policy. To mitigate the issue of model coverage, previous works (Yu et al., 2020; Kidambi et al., 2020) incorporate the ensemble-based uncertainty as a penalty into the reward function, preventing the agent from sampling within unsafe regions of $\hat{T}$.

### 2.3 ANY-STEP DYNAMICS MODEL

Lin et al. (2025) proposes the Any-step Dynamics Model (ADM) $\hat{T}(s_{t+k}, r_{t+k}|s_t, a_{t:t+k-1})$ to model the distribution of $s_{t+k}$ and $r_{t+k}$ after executing a $k$-step action sequence $a_{t:t+k-1} = (a_t, a_{t+1}, \cdots, a_{t+k-1})$ at $s_t$, where $k$ can be any integer under the limit of the maximum stride $m$. For future state prediction at each step, ADM has $m$ backtracking choices. For instance, $(s_t, a_t), (s_{t-1}, a_{t-1:t}), (s_{t-2}, a_{t-2:t}), \cdots, (s_{t-m+1}, a_{t-m+1:t})$ are all selectable as ADM's inputs to predict $s_{t+1}$. After reducing bootstrapping prediction to direct prediction, ADM can mitigate the compounding error in model roll-outs. What's more, the naturally diverse predictions from different backtracking choices can be utilized to estimate the model uncertainty, which has shown great consistency with the model error in ADMPO (Lin et al., 2025) and Whale (Zhang et al., 2024b).

### 3 METHOD

In this section, we design the second version of ADM (ADM-v2) to support full-horizon roll-outs and propose an efficient algorithm called PARoll to speed up any-step roll-out with uncertainty estimation. Furthermore, we introduce how to evaluate and learn the policy offline with ADM-v2 and PARoll.

### 3.1 ADM-V2

The architecture of ADM-v2 is clear and straightforward, as illustrated in Figure 1(b). We also illustrate ADM in Figure 1(a) for comparison. Following the definition of ADM (Lin et al., 2025), ADM-v2 can be denoted by $\hat{T}_\theta(s_{t+k}, r_{t+k}|s_t, a_{t:t+k-1})$, where $\theta$ represents the neural parameters. $k$ is the stride of the multi-step direct prediction, which can be any integer limited by the maximum prediction stride $m$, i.e., $1 \le k \le m$. The input of ADM-v2 is a state $s_t$ and a $k$-step action sequence $a_{t:t+k-1}$. The output is modeled as two independent Gaussian distributions of $s_{t+k}$ and $r_{t+k}$.

To handle the input action sequence with variable step sizes, we utilize an RNN (Elman, 1990) with a GRU (Cho et al., 2014) to implement ADM-v2, the same as ADM. The difference is that ADM-v2 decouples each recurrent forward of the GRU cell from the backtracked state $s_t$. Then the GRU cell inside ADM-v2 no longer takes $s_t$ as input repeatedly. There are three elements within ADM-v2, including a GRU $g_\theta$, a state encoder $\text{enc}_\theta$, and a transition decoder $\text{dec}_\theta$. Both the state encoder and transition decoder are implemented using MLPs (Goodfellow et al., 2016). As a whole, $g_\theta$, $\text{enc}_\theta$, and $\text{dec}_\theta$ constitute $\hat{T}_\theta$ together.

The backtracked state $s_t$ is encoded to a latent vector $h_t$ by the state encoder, i.e., $h_t = \text{enc}_\theta(s_t)$. We directly view $h_t$ as the initial hidden state of the GRU cell. With $h_t$ and $a_t$ as input, the hidden state is updated to $h_{t+1}$ by GRU, i.e., $h_{t+1} = g_\theta(h_t, a_t)$. Next, the transition decoder decodes $h_{t+1}$ to the distributions of $s_{t+1}$ and $r_{t+1}$. Concretely, $(\mu_{t+1}^s, \Sigma_{t+1}^s, \mu_{t+1}^r, \Sigma_{t+1}^r) = \text{dec}_\theta(h_{t+1})$, then $\mathcal{N}(\mu_{t+1}^s, \Sigma_{t+1}^s)$ and $\mathcal{N}(\mu_{t+1}^r, \Sigma_{t+1}^r)$ becomes the Gaussian distributions for $s_{t+1}$ and $r_{t+1}$

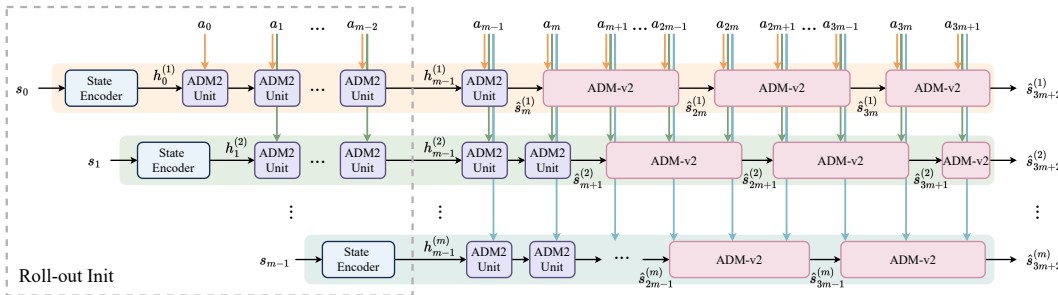

Figure 2: Illustration of the **P**arallel **A**ny-step **Roll**-out (PARoll), which is designed based on ADM-v2 and able to provide diverse predictions in parallel.

respectively. We regard the combination of GRU and transition decoder as an integrated ADM2 Unit. The hidden state $h_{t+1}$ can be continually propagated to the ADM2 unit to predict subsequent states and rewards, until the maximum stride $m$ is reached.

As an end-to-end neural network, ADM-v2 does not rely on any explicit supervision for intermediate variables, including the hidden state $h$. The joint distribution for each stride $k$ ($1 \leq k \leq m$) is optimized by directly maximizing the log likelihood

$$J_{\hat{T}}(\theta) = \frac{1}{m} \sum_{k=1}^{m} \mathbb{E}_{(s_t, a_{t:t+k-1}, r_{t+k}, s_{t+k}) \sim \mathcal{D}_{\text{env}}} \left[ \log \hat{T}_\theta(s_{t+k}, r_{t+k} | s_t, a_{t:t+k-1}) \right], \tag{1}$$

which is averaged among $m$ strides. The gradient of Equation equation 1 is directly backpropagated through the GRU, state encoder, and transition decoder.

ADM-v2 can improve future predictions by reducing bootstrapping prediction to multi-step direct prediction. For long-horizon prediction, we divide roll-outs into multiple windows of length $m$. The first window starts from the real state $s_0$, with the following $m$ states predicted without any error accumulation, and ends at the predicted $\hat{s}_m$. For the $n$-th window ($n > 1$), the starting state $\hat{s}_{(n-1)m}$ is encoded by the state encoder to $h_{(n-1)m}$. Along with the implicit update from $h_{(n-1)m}$ to $h_{nm}$ through $m$ GRU forward propagations, $\hat{s}_{(n-1)m+1}$ to $\hat{s}_{nm}$ are predicted without state bootstrapping. Therefore, the state bootstrapping only occurs when switching between windows. The roll-out process described above does not account for prediction diversity and uncertainty estimation. In the following subsection, we introduce the **P**arallel **A**ny-step **Roll**-out (PARoll) to address these limitations.

## 3.2 PARALLEL ANY-STEP ROLL-OUT

As introduced by ADM (Lin et al., 2025), one advantage of ADM is to generate diverse predictions and estimate the model uncertainty without an ensemble. However, ADM has to backtrack historical states from different distances and recursively call the GRU cell at each step, which costs unnecessary time consumption during roll-outs. The details are described in Appendix E.1.

ADM-v2 removes the mechanism of backtracking and rolls out diverse predictions in parallel, as shown by Figure 2, the illustration of PARoll. At the beginning of a roll-out, we sample a batch of state-action sequences of $m-1$ steps from $\mathcal{D}_{\text{env}}$, $(s_0, a_0, s_1, a_1, \cdots, s_{m-2}, a_{m-2}, s_{m-1})$. There are $m$ selectable states at different time steps, each corresponding to a timeline for the roll-out. For the first timeline, after encoding $s_0$ to $h_0^{(1)}$ and recursively feeding $(a_0, a_1, \cdots, a_{m-2})$ into ADM2 Unit, we obtain the first hidden state at step $m-1$, denoted as $h_{m-1}^{(1)}$, where the superscript indicates the index of the timeline and the subscript indicates the time step. Generally, for the $i$-th ($1 \leq i < m$) timeline, after encoding $s_{i-1}$ to $h_{i-1}^{(i)}$ and recursively feeding $(a_{i-1}, a_i, \cdots, a_{m-2})$ into ADM2 Unit, we obtain the $i$-th hidden state $h_{m-1}^{(i)}$ at step $m-1$. Specially, the $m$-th hidden state $h_{m-1}^{(m)}$ is directly obtained after encoding the $s_{m-1}$. After the process of roll-out initialization above, we obtain $m$ diverse starting hidden states $(h_{m-1}^{(1)}, h_{m-1}^{(2)}, \cdots, h_{m-1}^{(m)})$ at step $m-1$ for ADM-v2.

At the first step of the roll-out, the agent chooses an action $a_{m-1}$ according to $a_{m-1} \sim \pi(\cdot|s_{m-1})$. Then $m$ hidden states are in parallel updated by the ADM2 Unit after inputting $a_{m-1}$. For each $i \in \{1, 2, \cdots, m\}$, after updating the hidden state by $h_m^{(i)} = g_\theta(h_{m-1}^{(i)}, a_{m-1})$, we have

$$(\mu_{s_m}^{(i)}, \Sigma_{s_m}^{(i)}, \mu_{r_m}^{(i)}, \Sigma_{r_m}^{(i)}) = \text{dec}_\theta(h_m^{(i)}), \ \ \hat{s}_m^{(i)} \sim \mathcal{N}(\mu_{s_m}^{(i)}, \Sigma_{s_m}^{(i)}), \ \ \hat{r}_m^{(i)} \sim \mathcal{N}(\mu_{r_m}^{(i)}, \Sigma_{r_m}^{(i)}). \tag{2}$$

$(\hat{s}_m^{(1)}, \hat{s}_m^{(2)}, \cdots, \hat{s}_m^{(m)})$ are $m$ predictions for the state $s_m$ at time step $m$, transitioned from different backtracked states. Concretely, $\hat{s}_m^{(1)}$ is transitioned from $s_0$ after $(a_0, a_1, \cdots, a_{m-1})$, $\hat{s}_m^{(2)}$ is transitioned from $s_1$ after $(a_1, a_2, \cdots, a_{m-1})$, ......, and $\hat{s}_m^{(m)}$ is transitioned from $s_{m-1}$ after $a_{m-1}$. The discrepancy in transition strides yields diversity in predictions naturally.

For further roll-out, we repeat updating hidden states by $h_t^{(i)} = g_\theta(h_{t-1}^{(i)}, a_{t-1})$ and generating states by decoding $h_t^{(i)}$ to $\hat{s}_t^{(i)}$ for each $i$ at each step $t > m$. Both the hidden state updating and decoding operations are finished in parallel. Limited by the maximum stride $m$ set at the training phase of ADM-v2, the hidden state can't be propagated directly along the roll-out. We reset the hidden state when the direct prediction reaches the maximum transition stride for each timeline. For the $i$-th timeline, the starting state at the roll-out initialization is $s_{i-1}$, then reset time nodes between direct predictions are $(m + i - 1, 2m + i - 1, 3m + i - 1, \cdots)$. We reset hidden states at corresponding time nodes for each timeline by calling the state encoder to encode predicted states, *i.e.*, $h_{nm+i-1}^{(i)} = \text{enc}_\theta(\hat{s}_{nm+i-1}^{(i)})$ for the $n$-th stride. Practically, at each step $t$, only one timeline needs to reset the hidden state, the timeline $(t \bmod m) + 2$. Overall, each step only needs one state encoder forward, one parallel GRU cell forward, and one parallel transition decoder forward. The pseudo code of PARoll is presented in Appendix D.2. Each roll-out in ADM-v2 is full-horizon.

For each $t \geq m - 1$, we uniformly choose one from the diverse predictions $(\hat{s}_{t+1}^{(1)}, \hat{s}_{t+1}^{(2)}, \cdots, \hat{s}_{t+1}^{(m)})$ as the prediction and utilize them to estimate the model uncertainty (Yu et al., 2020; Kidambi et al., 2020; Lu et al., 2022; Lin et al., 2025) at time step $t$. For any maximum stride $m$ and the corresponding learned $\hat{T}_\theta$, the uncertainty of $\hat{T}_\theta$ at $(s_t, a_t)$ is quantified as

$$\mathcal{U}^{\text{ADM2}}(s_t, a_t) = \mathbb{E}\left[\left\|\text{Var}_{k \sim \text{Uniform}(m)}\left[\hat{s}_{t+1}^{(k)}\right]\right\|\right] = \mathbb{E}\left[\left\|\frac{1}{m}\sum_{k=1}^{m}\left((\Sigma_{s_t}^{(k)})^2 + (\mu_{s_t}^{(k)})^2\right) - (\bar{\mu}_t)^2\right\|_1\right], \tag{3}$$

where $\bar{\mu}_t = \frac{1}{m}\sum_{k=1}^{m}\mu_{s_t}^{(k)}$. The uncertainty term $\mathcal{U}^{\text{ADM2}}$ can be computed in parallel at each step by PARoll. We make further analysis for this any-step uncertainty in Section 3.3.

## 3.3 ADM-V2 FOR OFFLINE POLICY LEARNING

With a learned ADM-v2, offline policy exploration can directly happen in it. Unfortunately, the policy inevitably falls into some risky regions where the dynamics model is uncertain since the given dataset is limited. Model uncertainty estimation (Yu et al., 2020; Kidambi et al., 2020; Sun et al., 2023; Lin et al., 2025) can penalize actions that would lead the policy to regions not covered by the dynamics model. Our uncertainty term $\mathcal{U}^{\text{ADM2}}$ is estimated naturally during PARoll. Assuming $\mathcal{U}^{\text{ADM2}}$ is an admissible error estimator (Yu et al., 2020) whose numerical value at each state-action pair can bound the model error, we can find a penalty coefficient $\beta$ to make $\beta \cdot \mathcal{U}^{\text{ADM2}}$ become a valid $\xi$-uncertainty quantifier (Jin et al., 2021), as shown by Theorem 3.1.

**Theorem 3.1.** $\beta \cdot \mathcal{U}^{\text{ADM2}}$ *is a valid $\xi$-uncertainty quantifier. Specifically,*

$$\left|\hat{\mathcal{T}}^\pi Q(s_t, a_t) - \mathcal{T}^\pi Q(s_t, a_t)\right| \leq \beta \cdot \mathcal{U}^{\text{ADM2}}(s_t, a_t), \tag{4}$$

*where $\hat{\mathcal{T}}^\pi$ is the proxy operator induced by ADM-v2 to estimate the true Bellman operator $\mathcal{T}^\pi$.*

*Proof.* See Appendix C. $\qquad\square$

According to the sub-optimality theorem derived by Jin et al. (2021), any $\xi$-uncertainty quantifier as a penalty term during Q estimation (Sutton & Barto, 2018) can improve the policy $\pi$ to be close to

the optimal policy $\pi^\star$ with a bounded optimality gap. Penalizing the original Bellman operator with $\beta \cdot \mathcal{U}^{\text{ADM2}}$, we have

$$\hat{\mathcal{T}}^{\text{ADM2}}Q(s_t, a_t) := \hat{\mathcal{T}}^{\pi}Q(s_t, a_t) - \beta \cdot \mathcal{U}^{\text{ADM2}}(s_t, a_t). \tag{5}$$

Intuitively, for risky regions, larger model uncertainty corresponds to a looser bound on the Bellman error, thereby increasing the likelihood of Q-value overestimation. Penalizing Q estimation according to the magnitude of uncertainty helps prevent inflated Q-values in uncertain regions. This penalty does not interfere with policy learning on in-distribution samples, since their uncertainty is low and the penalty term remains inactive. Hence, the essence here is to identify a policy that achieves great performance within reliable state–action regions. The algorithm of policy learning in ADM-v2 via full-horizon roll-out is called ADM2PO-fh, with pseudo code shown in Appendix D.4.

## 3.4 ADM-V2 FOR OFF-POLICY EVALUATION

After learning an ADM-v2 $\hat{T}_\theta$ from a fixed dataset, we directly regard it as a surrogate environment. For any given policy, we can roll out several full-horizon trajectories in $\hat{T}_\theta$ and compute the average return. The pseudo code of off-policy evaluation (OPE) in ADM-v2 is shown in Appendix D.3.

There exists a gap between the evaluated return and the true return, whose magnitude depends on the accuracy of long-horizon roll-outs in $\hat{T}_\theta$. We characterize the relationship between this gap and the model error through Theorem 3.2.

**Theorem 3.2** (Performance Bound)**.** *Suppose the divergence of the $k$-step transition is bounded as*

$$\max_n \mathbb{E}_{s_{nm} \sim \rho^{\pi_D}} D_{\text{TV}}(\mathbb{P}(s_{nm+k}|s_{nm})\|\hat{\mathbb{P}}(s_{nm+k}|s_{nm})) \leq \delta^k$$

*for each $k \in \{1, 2, \cdots, m\}$, and $\delta_{\max} = \max_{k \in \{1,2,\cdots,m\}} \delta^k$. $\hat{\mathbb{P}}$ indicates the multi-step transition distribution induced by $\hat{T}$ and the policy $\pi$, and $\mathbb{P}$ indicates the multi-step transition distribution induced by the true dynamics $T$ and the policy $\pi$. The policy divergence is bounded as $\max_s D_{\text{TV}}(\pi_D(a|s)\|\pi(a|s)) \leq \epsilon_\pi$. Then the return gap is bounded as*

$$|\eta(\pi) - \hat{\eta}(\pi)| \leq \frac{2\gamma r_{\max}\delta_{\max}}{(1-\gamma)(1-\gamma^m)} + \frac{2\gamma r_{\max}\epsilon_\pi}{(1-\gamma)^2} + \frac{4r_{\max}\epsilon_\pi}{1-\gamma} = C(\delta_{\max}, \epsilon_\pi). \tag{6}$$

*Proof.* See Appendix C. $\square$

Theorem 3.2 states that the true performance of a policy can be bounded within a range of $\pm C(\delta_{\max}, \epsilon_\pi)$ around the performance evaluated by ADM-v2. The value of $C(\delta_{\max}, \epsilon_\pi)$ determines the reliability of ADM-v2 for OPE. If setting $m = 1$, the bound becomes $\frac{2\gamma r_{\max}\delta^1}{(1-\gamma)^2} + \frac{2\gamma r_{\max}\epsilon_\pi}{(1-\gamma)^2} + \frac{4r_{\max}\epsilon_\pi}{1-\gamma}$ and is equivalent with the performance bound (Janner et al., 2019) of the single-step dynamics model. Once the condition $\delta_{\max} < \frac{\delta^1(1-\gamma^m)}{1-\gamma}$ is satisfied, which is shown empirically in Section 4.1, we can assert that the bound of ADM-v2 is tighter than that of a single-step dynamics model.

## 4 EXPERIMENTS

Our experiments focus on these main questions: (1) How reliable are the trajectories rolled out by ADM-v2? (2) How effective is off-policy evaluation in ADM-v2? (3) How well is the performance after learning the policy in ADM-v2? (4) How well is the uncertainty estimator $\mathcal{U}^{\text{ADM2}}$? (5) How does the roll-out horizon affect the performance? (6) Does ADM-v2 learn better hidden representations?

## 4.1 DYNAMICS MODEL EVALUATION

We evaluate ADM-v2 in terms of prediction accuracy and efficiency, in comparison with four baseline dynamics models: (1) **ADM** (Lin et al., 2025): The original version of ADM, see Appendix E.1 for details; (2) **Ensemble Dynamics Model (EDM)** (Janner et al., 2019; Yu et al., 2020; Ni et al., 2025): EDM consists of several single-step dynamics models learned from different initial neural parameters

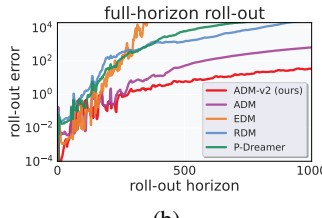 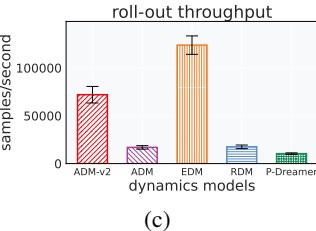

Figure 3: Evaluation of ADM-v2 and baselines on `hopper-medium-v2`, averaged over 5 seeds. (a) Comparison between direct prediction and bootstrapping prediction; (b) Compounding error (log scale) curves in full-horizon roll-out; (c) Roll-out throughput (number of samples per second).

and uniformly chooses one to make predictions at each step; (3) **Vanilla RNN Dynamics Model (RDM)**: The ablative dynamics model designed to eliminate the performance impact of the RNN itself. It inputs the sequence of state-action pairs instead of only the first state of a stride and the action sequence. See Appendix E.2 for details; (4) **Proprioceptive Dreamer (P-Dreamer)**: Dreamer (Hafner et al., 2020; 2021; 2023) focuses on the visual RL tasks. We modify the structure to adapt to the proprioceptive RL tasks where the observation is just a vector, see Appendix E.3 for details.

We conduct the evaluation on D4RL (Fu et al., 2020) MuJoCo datasets and demonstrate the complete results in Appendices G.1 to G.3. Figure 3 show the result on hopper-medium-v2 dataset. We divide the dataset into a training set to train dynamics models and an evaluation set to evaluate models.

The experiment includes three parts. The first part is the comparison between direct prediction and bootstrapping prediction, as shown by Figure 3(a). We sample 100 sequences from the evaluation set and calculate the $m$-step direct prediction error of ADM-v2 and ADM, the $m$-step bootstrapping prediction error of EDM, and $\frac{\delta^1(1-\gamma^m)}{1-\gamma}$, for each $m$ from 1 to 8. Comparing the curve of ADM-v2 with $\frac{\delta^1(1-\gamma^m)}{1-\gamma}$, we can find a proper $m$ to satisfy $\delta_{\max} = \max_{k \in \{1,2,\cdots,m\}} \delta^k < \frac{\delta^1(1-\gamma^m)}{1-\gamma}$. While setting $m$ as the proper value, the performance bound of ADM-v2 is tighter than that of single-step dynamics models like EDM. It is also observed that the direct prediction error curves of ADM-v2 and ADM are both below the bootstrapping prediction error curve of EDM, indicating the advantage of direct prediction in improving future predictions. Furthermore, the direct prediction error of ADM-v2 is smaller than that of ADM, showing the structural advancement of ADM-v2 over ADM.

The second part is the evaluation of the compounding error in full-length roll-outs, as shown by Figure 3(b). We sample 100 trajectories from the evaluation set and execute actions sequentially in each trained model to calculate the average roll-out error at each step. Since the maximum episode length in MuJoCo (Todorov et al., 2012) tasks is 1000, we set the maximum roll-out horizon to 1000 as well. We observe that the roll-out error curve of ADM-v2 is below the curve of other baselines in most cases. Until the episode end, the roll-out error of ADM-v2 still does not tend toward infinity.

The third part is the evaluation of the roll-out throughput, which is calculated as the average number of roll-out samples per second, measuring the time efficiency of the model roll-out. For fairness, the roll-out batch size of each dynamics model is set to be the same. As shown by Figure 3(c), the roll-out throughput of ADM-v2 is greater than other RNN-based dynamics models, ADM, RDM, and P-Dreamer. EDM achieves high time efficiency in roll-out due to its simple architecture. However, the disadvantage of ADM-v2 compared to EDM is not significant. Considering the fidelity advantage of ADM-v2 in full-horizon roll-outs, this slight cost in efficiency is acceptable.

## 4.2 OFF-POLICY EVALUATION

We conduct the Off-policy Evaluation (OPE) on the DOPE (Fu et al., 2021) benchmark and use three metrics, *i.e.*, normalized absolute error, rank correlation, and regret@1, to evaluate the consistency between the estimated returns and the true returns of given policies. The detailed definition of these metrics is described in Appendix G.4. We let the policy make full-horizon roll-outs in EDM, RDM, P-Dreamer, ADM, and ADM-v2 to estimate the return and calculate the metrics, respectively. Besides, we compare ADM-v2 for OPE with 5 model-free algorithms, including Fitted Q-Evaluation (FQE)

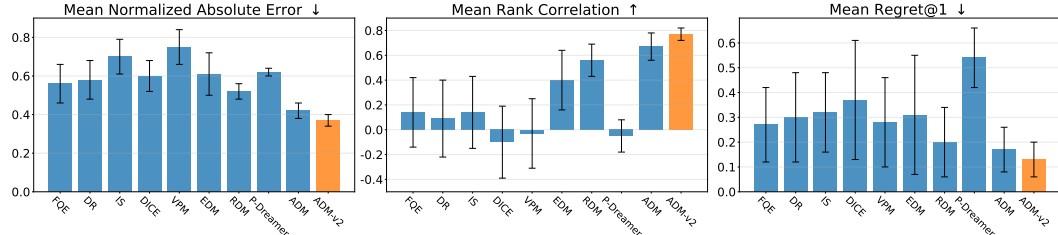

Figure 4: The overall OPE results across 15 DOPE tasks.

(Le et al., 2019), Doubly Robust OPE (DR) (Jiang & Li, 2016), Importance Sampling (IS) (Kostrikov & Nachum, 2020), DICE (Yang et al., 2020), and Variational Power Method (VPM) (Wen et al., 2020). We select 15 datasets from Hopper, HalfCheetah, and Walker3d in the DOPE benchmark, each with five dataset types. The overall OPE results across these 15 DOPE tasks are shown in Figure 4, averaged over 3 seeds. The exhaustive experimental data can be found in Appendix G.4.

In terms of all three metrics, ADM-v2 for OPE significantly outperforms the five model-free OPE algorithms and also surpasses the performance of other dynamics models for OPE. This result demonstrates the advantages of ADM-v2 for OPE and, at the same time, indirectly highlights its strong predictive capability as a dynamics model.

## 4.3 OFFLINE RL PERFORMANCE

Table 1: Normalized scores after offline learning on D4RL MuJoCo tasks.

| Task Name | Model-free | | Model-based (short branched roll-out) | | | | Model-based (full-horizon roll-out) | | | | |
|---|---|---|---|---|---|---|---|---|---|---|---|
| | CQL | EDAC | MOPO | MOBILE | MOREC | ADMPO | MOPO-fh | RMPO-fh | PDPO-fh | ADMPO-fh | ADM2PO-fh (ours) |
| hopper-rand | 5.3 | 25.3 | 31.7 | 31.9 | 32.1 | 32.7 | 31.7 | 33.3 | 0.9 | 20.9 | **42.7 ± 3.4** |
| halfcheetah-rand | 31.3 | 28.4 | 38.5 | 39.3 | **51.6** | 45.4 | 26.6 | **51.1** | 5.7 | 27.0 | 45.9 ± 3.9 |
| walker2d-rand | 5.4 | 16.6 | 7.4 | 17.9 | 23.5 | 22.2 | 2.2 | **27.0** | 0.0 | 0.0 | **27.0 ± 2.5** |
| hopper-med | 61.9 | 101.6 | 62.8 | 106.6 | **107.0** | 107.4 | 104.2 | 52.1 | 13.6 | 64.8 | **107.6 ± 0.3** |
| halfcheetah-med | 46.9 | 65.9 | 73.0 | 74.6 | 82.3 | 72.2 | -1.0 | 62.4 | -0.4 | 62.7 | **85.4 ± 3.8** |
| walker2d-med | 79.5 | 92.5 | 84.1 | 87.7 | 89.9 | 93.2 | 27.0 | 8.9 | -0.2 | 58.7 | **111.6 ± 4.8** |
| hopper-med-rep | 86.3 | 101.0 | 103.5 | 103.9 | 105.1 | 104.4 | 94.2 | 78.0 | 22.9 | 83.4 | **107.0 ± 0.5** |
| halfcheetah-med-rep | 45.3 | 61.3 | 72.1 | 71.7 | **76.5** | 67.6 | 0.82 | 62.3 | 7.6 | 49.0 | 67.5 ± 2.7 |
| walker2d-med-rep | 76.8 | 87.1 | 85.6 | 89.9 | 95.5 | 95.6 | 11.4 | 6.6 | -0.3 | 26.1 | **106.3 ± 4.0** |
| hopper-med-exp | 96.9 | 110.7 | 81.6 | 112.6 | **113.3** | 112.7 | 11.5 | 111.3 | 2.0 | 110.2 | 111.7 ± 0.4 |
| halfcheetah-med-exp | 95.0 | 106.3 | 90.8 | 108.2 | 112.1 | 103.7 | 28.6 | 77.5 | 1.0 | 90.7 | **113.2 ± 1.4** |
| walker2d-med-exp | 109.1 | 114.7 | 112.9 | 115.2 | 115.8 | 114.9 | 100.2 | 52.7 | 0.6 | 83.8 | **125.1 ± 0.3** |
| Average | 61.6 | 76.0 | 70.3 | 80.0 | 83.7 | 81.0 | 36.5 | 51.9 | 4.5 | 56.4 | **87.6 ± 1.3** |

Table 2: Normalized scores after offline learning on NeoRL MuJoCo tasks.

| Task Name | Model-free | | Model-based (short branched roll-out) | | | | Model-based (full-horizon roll-out) | | | | |
|---|---|---|---|---|---|---|---|---|---|---|---|
| | CQL | EDAC | MOPO | MOBILE | MOREC | ADMPO | MOPO-fh | RMPO-fh | PDPO-fh | ADMPO-fh | ADM2PO-fh (ours) |
| neorl-hopper-L | 16.0 | 18.3 | 6.2 | 17.4 | **53.5** | 22.3 | 24.8 | 11.2 | 15.3 | 17.3 | 32.3 ± 3.5 |
| neorl-halfcheetah-L | 38.2 | 31.3 | 40.1 | 54.7 | 25.4 | 52.8 | 25.4 | 54.6 | 20.0 | 49.1 | **61.7 ± 2.4** |
| neorl-walker2d-L | 44.7 | 40.2 | 11.6 | 37.6 | 65.0 | 55.9 | 5.2 | 68.6 | 30.4 | 16.6 | **76.3 ± 3.2** |
| neorl-hopper-M | 64.5 | 44.9 | 1.0 | 51.1 | 84.1 | 51.5 | 70.4 | 68.9 | 8.4 | 59.2 | **105.2 ± 0.9** |
| neorl-halfcheetah-M | 54.6 | 54.9 | 62.3 | 77.8 | **83.5** | 69.3 | 37.7 | 76.3 | 13.4 | 60.9 | 78.1 ± 3.1 |
| neorl-walker2d-M | 57.3 | 57.6 | 39.9 | 62.2 | 76.6 | 70.1 | 29.9 | 70.8 | 10.6 | 43.4 | **81.8 ± 2.3** |
| neorl-hopper-H | 76.6 | 52.5 | 11.5 | 87.8 | 90.3 | 87.6 | 79.7 | 60.4 | 8.4 | 38.2 | **106.8 ± 1.1** |
| neorl-halfcheetah-H | 77.4 | 81.4 | 65.9 | 83.0 | 72.8 | 84.0 | 0.2 | 79.2 | 2.7 | 72.8 | **85.7 ± 5.6** |
| neorl-walker2d-H | 75.3 | 75.5 | 18.0 | 74.9 | **83.0** | 82.2 | 18.1 | 33.3 | 46.4 | 72.9 | 83.1 ± 1.9 |
| Average | 56.1 | 50.7 | 28.5 | 60.7 | 70.3 | 64.0 | 32.4 | 58.1 | 17.3 | 47.8 | **79.0 ± 1.8** |

We evaluate the policy performance of ADM2PO-fh (full-horizon roll-out in ADM2 for policy optimization) on the D4RL (Fu et al., 2020) benchmark and the NeoRL (Qin et al., 2022) benchmark. We select 12 D4RL MuJoCo tasks, *i.e.*, random (rand), medium (med), medium-replay (med-rep), and medium-expert (med-exp) datasets from Hopper, HalfCheetah, and Walker2d, and 9 NeoRL MuJoCo tasks, *i.e.*, low (L), medium (M), and high (H) levels of datasets from Hopper, HalfCheetah, and Walker2d. We compare the performance of offline RL in ADM-v2 with 2 model-free algorithms, CQL (Kumar et al., 2020) and EDAC (An et al., 2021), and 4 model-based algorithms based on the short branched model roll-out (Janner et al., 2019), MOPO (Yu et al., 2020), MOBILE (Sun et al., 2023),

MOREC (Luo et al., 2024), and ADMPO (Lin et al., 2025). Besides, we also make comparisons with the RL performance by making full-horizon roll-out in EDM, RDM, P-Dreamer, and ADM, corresponding to MOPO-fh (EDM), RMPO-fh (RDM), PDPO-fh (P-Dreamer), and ADMPO-fh (ADM). Results averaged over 5 seeds on D4RL and NeoRL are shown in Table 1 and Table 2.

ADM2PO-fh achieves a significant advantage in policy performance over previous offline RL baselines. In particular, compared to the prior state-of-the-art MOREC, we obtain more than 4.6% performance gain on D4RL and over 12.8% gain on NeoRL. Moreover, among all models, only ADM-v2 can enable the policy to achieve strong performance by making full-horizon roll-outs. Notably, MOPO and ADMPO both suffer from severe performance degradation when conducting full-horizon roll-outs (MOPO-fh and ADMPO-fh), since they lack the capability for long-horizon predictions. The results of ADM2PO-fh demonstrate that when a dynamics model possesses the ability for full-horizon roll-outs, it can lead to substantial improvements in policy performance. We further study the influence of roll-out horizon in Appendix G.6.

### 4.4 UNCERTAINTY QUANTIFICATION

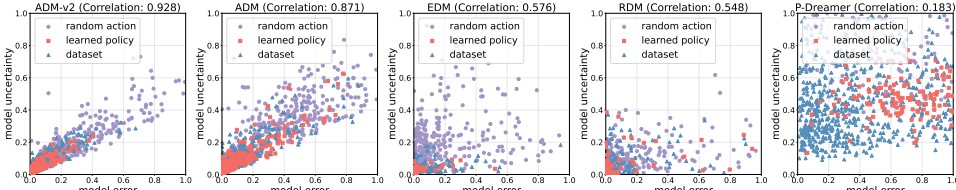

Figure 5: Comparison of uncertainty quantification among models on `hopper-medium-v2`.

We sample data points in ADM-v2, ADM, EDM, RDM, and P-Dreamer by random action selection, the offline-learned policy, and the dataset behavior. For each sample, we compute the model error and model uncertainty, and scatter the results on the hopper-medium-v2 dataset in Figure 5. Results on more datasets are shown in Appendix G.11. ADM-v2 demonstrates reliable uncertainty quantification, as samples with larger model errors tend to exhibit correspondingly larger model uncertainties. Notably, ADM-v2 achieves a correlation coefficient of 0.928 between model uncertainty and model error, outperforming all other models. This indicates that $\mathcal{U}^{\mathrm{ADM2}}$ satisfies the assumption of an admissible error estimator (Yu et al., 2020). Moreover, ADM-v2 can distinguish different policies better. As samples of the random policy exhibit a clear distributional shift from the hopper-medium-v2 dataset, their uncertainties should be higher than the dataset behavior in expectation. Conversely, the uncertainty penalty drives the learned policy toward the safe regions covered by the dataset, where model uncertainty remains lower than the dataset behavior. The scatter plots of ADM-v2 reveal these phenomena more clearly than other models.

### 4.5 STUDY ON ROLL-OUT HORIZON

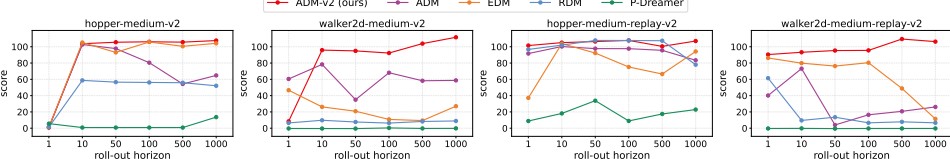

Figure 6: Normalized scores under different roll-out horizons on four D4RL tasks.

To examine the effect of roll-out horizon, we measure the normalized scores obtained by policy optimization in five dynamics models on hopper-medium, walker2d-medium, hopper-medium-replay, and walker2d-medium-replay tasks under roll-out horizons of {10, 50, 100, 500, 1000} (Figure 6). We find that only ADM-v2 exhibits consistent performance improvements as the roll-out horizon increases, while most of the remaining methods either suffer performance degradation or maintain low scores as the horizon grows, especially on the difficult walker2d-medium task. In particular, ADM achieves strong performance under short roll-outs but deteriorates rapidly as the roll-out horizon becomes longer. These results suggest that, compared to other models, ADM-v2 can effectively

leverage long-horizon roll-outs for better policy learning, owing to its high-fidelity future state predictions.

### 4.6 STUDY ON HIDDEN REPRESENTATIONS

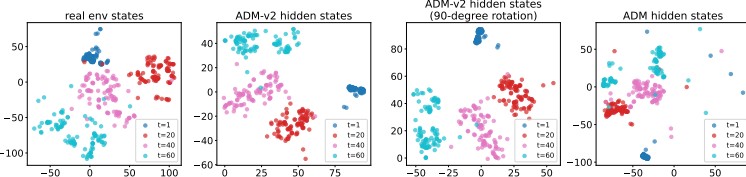

Figure 7: Visualizations of real environmental states and hidden representations in ADM-v2 and ADM, after t-SNE (Maaten & Hinton, 2008), on the `hopper-medium-v2` task.

We select the hopper-medium task and roll out multiple trajectories from the initial state in the real environment, ADM-v2, and ADM, using the action sequence of the dataset. During the roll-out, we record the real environmental states, the hidden states of ADM-v2, and the hidden states of ADM, and then apply t-SNE (Maaten & Hinton, 2008) for dimension reduction. We visualize the samples at time steps t = 1, 20, 40, 60, and the results are shown in Figure 7. Since all trajectories start from similar initial states, the samples at these four time steps should form four clusters in geometric space. The first plot in Figure 7 shows the visualization of real environmental states, where the geometric separation of the four clusters is clear. The second plot shows the hidden states of ADM-v2, which also exhibit well-defined cluster boundaries. In contrast, the fourth plot shows the hidden states of ADM, whose geometric structure appears significantly more entangled. Moreover, by rotating the ADM-v2 hidden-state embeddings by 90 degrees (shown in the third plot of Figure 7), we observe that the spatial structure of ADM-v2's hidden states aligns with that of the environmental states.

## 5 RELATED WORK

This work is related to offline MBRL and OPE. See Appendix H for detailed related work.

MBRL has shown strong potential in the offline setting, as it can leverage learned dynamics models to expand datasets and improve data efficiency. To mitigate the issue of accumulated roll-out error, prior works introduce conservative mechanisms and restrict the horizon of roll-outs, including MOPO (Yu et al., 2020), MOReL (Kidambi et al., 2020), MOBILE (Sun et al., 2023), MOREC (Luo et al., 2024), and ADMPO (Lin et al., 2025). In contrast, ADM-v2 introduces a new architecture that enables roll-outs across the full episode horizon, supporting more effective exploration and policy learning. ADM-v2 is an efficient and effective extension of ADM (Lin et al., 2025), which generalizes the direct state transition after executing a multi-step action sequence (Asadi et al., 2018; 2019; Che et al., 2018; Machado et al., 2023) into an any-step formulation. What's more, multi-step predictions are also considered by Amos et al. (2018); Hafner et al. (2019).

OPE methods aim to estimate the value of a target policy using only pre-collected data. They are typically categorized into two types. One type is model-free methods like fitted Q-evaluation (Le et al., 2019), importance sampling (IS) (Kostrikov & Nachum, 2020), the doubly robust method (Jiang & Li, 2016), and DICE (Yang et al., 2020). Another type is model-based methods (Chen et al., 2024b;a). Our approach introduces a new dynamics model that proves effective for OPE.

## 6 CONCLUSION

In this work, we address the long-standing challenge of enabling reliable full-horizon roll-outs in offline MBRL, where existing dynamics models typically suffer from severe error accumulation. We propose ADM-v2 to substantially improve the long-horizon prediction accuracy, enabling both policy evaluation and policy optimization directly with full-horizon trajectories. Experiments on D4RL, NeoRL, and DOPE benchmarks demonstrate that ADM-v2 achieves state-of-the-art performance, surpassing prior model-based and model-free baselines, with more than 4.6% gain on D4RL and 12.8% gain on NeoRL. Our results highlight the importance of reliable full-horizon roll-outs and establish ADM-v2 as an effective dynamics model for offline reinforcement learning.

ACKNOWLEDGMENTS

This work was supported by the Yangtze River Delta Science and Technology Innovation Community Joint Research Program 2024CSJZN00302, the National Science Foundation of China (62495093,62495090), and the Natural Science Foundation of Jiangsu (BK20243039). The authors thank anonymous reviewers for their helpful discussions and suggestions for improving the article.

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

## A  THE USE OF LARGE LANGUAGE MODELS (LLMS)

This paper employs LLMs solely for refining the content of Sections 1 and 3, focusing on improving clarity, coherence, and conciseness of the presentation.

## B  USEFUL LEMMAS

**Lemma B.1** (TV Divergence of Joint Distributions). *Given two distributions $\mathbb{P}_1(x, y) = \mathbb{P}_1(x)\mathbb{P}_1(y|x)$ and $\mathbb{P}_2(x, y) = \mathbb{P}_2(x)\mathbb{P}_2(y|x)$. We can bound the total variation divergence between $\mathbb{P}_1(x, y)$ and $\mathbb{P}_2(x, y)$ as*

$$D_{\mathrm{TV}}\left(\mathbb{P}_1(x, y)\|\mathbb{P}_2(x, y)\right) \leq D_{\mathrm{TV}}\left(\mathbb{P}_1(x)\|\mathbb{P}_2(x)\right) + \max_x D_{\mathrm{TV}}\left(\mathbb{P}_1(y|x)\|\mathbb{P}_2(y|x)\right). \quad (7)$$

*Proof.*

$$
\begin{aligned}
&D_{\mathrm{TV}}\left(\mathbb{P}_1(x, y)\|\mathbb{P}_2(x, y)\right) \\
=&\frac{1}{2}\sum_{x,y}|\mathbb{P}_1(x, y) - \mathbb{P}_2(x, y)| \\
=&\frac{1}{2}\sum_{x,y}|\mathbb{P}_1(x)\mathbb{P}_1(y|x) - \mathbb{P}_2(x)\mathbb{P}_2(y|x)| \\
=&\frac{1}{2}\sum_{x,y}|\mathbb{P}_1(x)\mathbb{P}_1(y|x) - \mathbb{P}_1(x)\mathbb{P}_2(y|x) + \left(\mathbb{P}_1(x) - \mathbb{P}_2(x)\right)\mathbb{P}_2(y|x)| \\
\leq&\frac{1}{2}\sum_{x,y}\mathbb{P}_1(x)\left|\mathbb{P}_1(y|x) - \mathbb{P}_2(y|x)\right| + \left|\mathbb{P}_1(x) - \mathbb{P}_2(x)\right|\mathbb{P}_2(y|x) \\
=&\frac{1}{2}\sum_{x,y}\mathbb{P}_1(x)\left|\mathbb{P}_1(y|x) - \mathbb{P}_2(y|x)\right| + \frac{1}{2}\sum_x|\mathbb{P}_1(x) - \mathbb{P}_2(x)| \\
=&E_{x\sim\mathbb{P}_1}\left[D_{\mathrm{TV}}\left(\mathbb{P}_1(y|x)\|\mathbb{P}_2(y|x)\right)\right] + D_{\mathrm{TV}}\left(\mathbb{P}_1(x)\|\mathbb{P}_2(x)\right) \\
\leq&\max_x D_{\mathrm{TV}}\left(\mathbb{P}_1(y|x)\|\mathbb{P}_2(y|x)\right) + D_{\mathrm{TV}}\left(\mathbb{P}_1(x)\|\mathbb{P}_2(x)\right).
\end{aligned}
\quad (8)
$$

$\square$

**Lemma B.2** (TV Divergence Bound of Rolled-out State Distributions). *Suppose the expected TV distance between two $k$-step transition distributions is bounded as*

$$\max_n \mathbb{E}_{s_{nm}\sim\mathbb{P}_1} D_{\mathrm{TV}}(\mathbb{P}_1(s_{nm+k}|s_{nm})\|\mathbb{P}_2(s_{nm+k}|s_{nm})) \leq \delta^k$$

*for each $k \in \{1, 2, \cdots, m\}$, and the initial state distributions are the same, $\mathbb{P}_1(s_0) = \mathbb{P}_2(s_0)$. Then the TV divergence between two state distributions at the end of the $n$-th stride is bounded as*

$$D_{\mathrm{TV}}(\mathbb{P}_1(s_{nm})\|\mathbb{P}_2(s_{nm})) \leq n\delta^m, \quad (9)$$

*and the TV divergence between two distributions of the $k$-th state in the $(n+1)$-th stride is bounded as*

$$D_{\mathrm{TV}}(\mathbb{P}_1(s_{nm+k})\|\mathbb{P}_2(s_{nm+k})) \leq n\delta^m + \delta^k. \quad (10)$$

(This lemma only considers the first timeline starting from $s_0$, the bound for other timelines of PARoll can be proved similarly.)

*Proof.* We first extend the absolute difference of $\mathbb{P}_1(s_{nm})$ and $\mathbb{P}_2(s_{nm})$,

$$
\begin{aligned}
&\left|\mathbb{P}_1(s_{nm}) - \mathbb{P}_2(s_{nm})\right| \\
&= \left| \sum_{s_{(n-1)m}} \mathbb{P}_1(s_{nm}|s_{(n-1)m})\mathbb{P}_1(s_{(n-1)m}) - \mathbb{P}_2(s_{nm}|s_{(n-1)m})\mathbb{P}_2(s_{(n-1)m}) \right| \\
&\le \sum_{s_{(n-1)m}} \left|\mathbb{P}_1(s_{nm}|s_{(n-1)m})\mathbb{P}_1(s_{(n-1)m}) - \mathbb{P}_2(s_{nm}|s_{(n-1)m})\mathbb{P}_2(s_{(n-1)m})\right| \\
&= \sum_{s_{(n-1)m}} \left| \left(\mathbb{P}_1(s_{nm}|s_{(n-1)m}) - \mathbb{P}_2(s_{nm}|s_{(n-1)m})\right)\mathbb{P}_1(s_{(n-1)m}) \right. \\
&\qquad\qquad \left. + \mathbb{P}_2(s_{nm}|s_{(n-1)m})\left(\mathbb{P}_1(s_{(n-1)m}) - \mathbb{P}_2(s_{(n-1)m})\right) \right| \\
&\le \sum_{s_{(n-1)m}} \mathbb{P}_1(s_{(n-1)m})\left|\mathbb{P}_1(s_{nm}|s_{(n-1)m}) - \mathbb{P}_2(s_{nm}|s_{(n-1)m})\right| \\
&\qquad\qquad + \mathbb{P}_2(s_{nm}|s_{(n-1)m})\left|\mathbb{P}_1(s_{(n-1)m}) - \mathbb{P}_2(s_{(n-1)m})\right|.
\end{aligned}
\tag{11}
$$

Next, we have

$$
\begin{aligned}
\epsilon_n &= D_{\mathrm{TV}}(\mathbb{P}_1(s_{nm})\|\mathbb{P}_2(s_{nm})) \\
&= \frac{1}{2}\sum_{s_{nm}} |\mathbb{P}_1(s_{nm}) - \mathbb{P}_2(s_{nm})| \\
&\le \frac{1}{2}\sum_{s_{nm}} \mathbb{E}_{s_{(n-1)m}\sim\mathbb{P}_1}\left[\left|\mathbb{P}_1(s_{nm}|s_{(n-1)m}) - \mathbb{P}_2(s_{nm}|s_{(n-1)m})\right|\right] \\
&\qquad + \frac{1}{2}\sum_{s_{nm}}\sum_{s_{(n-1)m}} \mathbb{P}_2(s_{nm}|s_{(n-1)m})\left|\mathbb{P}_1(s_{(n-1)m}) - \mathbb{P}_2(s_{(n-1)m})\right| \\
&= \frac{1}{2}\mathbb{E}_{s_{(n-1)m}\sim\mathbb{P}_1}\left[\sum_{s_{nm}}\left|\mathbb{P}_1(s_{nm}|s_{(n-1)m}) - \mathbb{P}_2(s_{nm}|s_{(n-1)m})\right|\right] \\
&\qquad + \frac{1}{2}\sum_{s_{(n-1)m}}\left|\mathbb{P}_1(s_{(n-1)m}) - \mathbb{P}_2(s_{(n-1)m})\right| \\
&= \delta_n^m + D_{\mathrm{TV}}(\mathbb{P}_1(s_{(n-1)m})\|\mathbb{P}_2(s_{(n-1)m})) \\
&= \delta_n^m + \epsilon_{n-1} \\
&= \sum_{i=1}^{n}\delta_i^m \le n\delta^m,
\end{aligned}
\tag{12}
$$

where we define

$$
\delta_n^k = \frac{1}{2}\mathbb{E}_{s_{(n-1)m}\sim\mathbb{P}_1}\left[\sum_{s_{(n-1)m+k}}\left|\mathbb{P}_1(s_{(n-1)m+k}|s_{(n-1)m}) - \mathbb{P}_2(s_{(n-1)m+k}|s_{(n-1)m})\right|\right],
$$

which is upper bounded by $\delta^k$. Similarly, we can prove

$$
D_{\mathrm{TV}}(\mathbb{P}_1(s_{nm+k})\|\mathbb{P}_2(s_{nm+k})) \le \delta_{n+1}^k + D_{\mathrm{TV}}(\mathbb{P}_1(s_{nm})\|\mathbb{P}_2(s_{nm})) \le n\delta^m + \delta^k
\tag{13}
$$

for each $k \in \{1, 2, \cdots, m\}$. $\qquad\square$

**Lemma B.3** (Bound of Return Gap). *Define the return of a given policy $\pi$ as*

$$
\eta = \mathbb{E}_{\rho^\pi}\left[\sum_{t=0}^{\infty}\gamma^t R(s_t, a_t)\right],
\tag{14}
$$

*where $R$ is the given reward function and $\max_{s,a} R(s,a) = r_{\max}$. The maximum TV divergence of the multi-step transition distributions is denoted as*

$$
\delta_{\max} = \max_{k\in\{1,2,\cdots,m\}}\delta^k.
\tag{15}
$$

*We use $\eta_1$ to denote the return of $\pi_1$ under transition $T_1$ and $\eta_2$ to denote the return of $\pi_2$ under transition $T_2$. $\mathbb{P}_1$ denotes the state-action (state) distribution induced by $\pi_1$ and $T_1$, and $\mathbb{P}_2$ denotes the state-action (state) distribution induced by $\pi_2$ and $T_2$. Suppose the expected TV distance between $\pi_1$ and $\pi_2$ satisfies*

$$\max_s D_{\mathrm{TV}}(\pi_1(a|s)\|\pi_2(a|s)) \le \epsilon_\pi, \tag{16}$$

*the return gap is bounded as*

$$|\eta_1 - \eta_2| \le \frac{2\gamma r_{\max}\delta_{\max}}{(1-\gamma)(1-\gamma^m)} + \frac{2r_{\max}\epsilon_\pi}{1-\gamma}. \tag{17}$$

*Proof.* First, we consider the expected return of the former $nm$ steps, which is defined as

$$\eta^{nm} = \mathbb{E}_{\rho^\pi}\left[\sum_{t=0}^{nm}\gamma^t R(s_t, a_t)\right]. \tag{18}$$

That is to say $\eta = \lim_{n\to\infty}\eta^{nm}$. The gap between $\eta_1^{nm}$ and $\eta_2^{nm}$ satisfies

$$\begin{aligned}
&|\eta_1^{nm} - \eta_2^{nm}| \\
&= \left|\eta_1^{(n-1)m} - \eta_2^{(n-1)m}\right| + \sum_{t=(n-1)m+1}^{nm}\left(\sum_{s_t,a_t}|\mathbb{P}_1(s_t,a_t) - \mathbb{P}_2(s_t,a_t)|\gamma^t R(s_t,a_t)\right) \\
&\le \left|\eta_1^{(n-1)m} - \eta_2^{(n-1)m}\right| + 2\sum_{t=(n-1)m+1}^{nm}\gamma^t r_{\max}D_{\mathrm{TV}}(\mathbb{P}_1(s_t,a_t)\|\mathbb{P}_2(s_t,a_t)).
\end{aligned} \tag{19}$$

By Lemma B.1, we have

$$\begin{aligned}
&D_{\mathrm{TV}}(\mathbb{P}_1(s_t,a_t)\|\mathbb{P}_2(s_t,a_t)) \\
&\le D_{\mathrm{TV}}(\mathbb{P}_1(s_t)\|\mathbb{P}_2(s_t)) + \max_{s_t}D_{\mathrm{TV}}(\pi_1(a_t|s_t)\|\pi_2(a_t|s_t)) \\
&\le D_{\mathrm{TV}}(\mathbb{P}_1(s_t)\|\mathbb{P}_2(s_t)) + \epsilon_\pi.
\end{aligned} \tag{20}$$

Hence, after applying the result of Lemma B.2, we have

$$\begin{aligned}
&|\eta_1^{nm} - \eta_2^{nm}| \\
&\le \left|\eta_1^{(n-1)m} - \eta_2^{(n-1)m}\right| + 2\gamma^{(n-1)m}\sum_{k=1}^m\gamma^k r_{\max}\left((n-1)\delta^m + \delta^k + \epsilon_\pi\right) \\
&\le \left|\eta_1^{(n-1)m} - \eta_2^{(n-1)m}\right| + 2\gamma^{(n-1)m}\left(n\delta_{\max} + \epsilon_\pi\right)\sum_{k=1}^m\gamma^k r_{\max} \\
&\le \left|\eta_1^0 - \eta_2^0\right| + 2\sum_{i=1}^n\gamma^{(i-1)m}\left(i\delta_{\max} + \epsilon_\pi\right)\sum_{k=1}^m\gamma^k r_{\max} \\
&\le 2\epsilon_\pi r_{\max} + 2\sum_{i=1}^n\gamma^{(i-1)m}\left(i\delta_{\max} + \epsilon_\pi\right)\sum_{k=1}^m\gamma^k r_{\max} \\
&= 2r_{\max}\left(\epsilon_\pi + \delta_{\max}\sum_{i=1}^n i\gamma^{(i-1)m}\sum_{k=1}^m\gamma^k + \epsilon_\pi\sum_{t=1}^{nm}\gamma^t\right) \\
&= 2r_{\max}\left(\epsilon_\pi + \delta_{\max}\sum_{i=1}^n i\gamma^{(i-1)m}\frac{\gamma(1-\gamma^m)}{1-\gamma} + \epsilon_\pi\sum_{t=1}^{nm}\gamma^t\right) \\
&= 2r_{\max}\left(\epsilon_\pi + \delta_{\max}\frac{1-\gamma^m}{\gamma^{m-1}(1-\gamma)}\sum_{i=1}^n i\gamma^{im} + \epsilon_\pi\sum_{t=1}^{nm}\gamma^t\right).
\end{aligned} \tag{21}$$

Finally, while $n$ tends to be $\infty$, we have

$$
\begin{aligned}
|\eta_1 - \eta_2| &= \lim_{n \to \infty} |\eta_1^{nm} - \eta_2^{nm}| \\
&\leq 2r_{\max} \left( \epsilon_\pi + \delta_{\max} \frac{1 - \gamma^m}{\gamma^{m-1}(1 - \gamma)} \sum_{i=1}^{\infty} i\gamma^{im} + \epsilon_\pi \sum_{t=1}^{\infty} \gamma^t \right) \\
&= 2r_{\max} \left( \frac{\gamma \delta_{\max}}{(1 - \gamma)(1 - \gamma^m)} + \frac{\epsilon_\pi}{1 - \gamma} \right) \\
&= \frac{2\gamma r_{\max} \delta_{\max}}{(1 - \gamma)(1 - \gamma^m)} + \frac{2r_{\max} \epsilon_\pi}{1 - \gamma}.
\end{aligned}
\tag{22}
$$

$\square$

**Lemma B.4** (Bound of Return Gap without Model Error). *Not considering the model error, we have*

$$
\max_n \mathbb{E}_{s_{nm} \sim \mathbb{P}_1} D_{\mathrm{TV}}(\mathbb{P}_1(s_{nm+k}|s_{nm}) \| \mathbb{P}_2(s_{nm+k}|s_{nm})) \leq k\epsilon_m = \delta^k.
\tag{23}
$$

*We use $\eta_1$ to denote the return of $\pi_1$ and $\eta_2$ to denote the return of $\pi_2$, under the same dynamics. Suppose the expected TV distance between $\pi_1$ and $\pi_2$ satisfies*

$$
\max_s D_{\mathrm{TV}}(\pi_1(a|s) \| \pi_2(a|s)) \leq \epsilon_\pi,
\tag{24}
$$

*the return gap is bounded as*

$$
|\eta_1 - \eta_2| \leq \frac{2\gamma r_{\max} \epsilon_\pi}{(1 - \gamma)^2} + \frac{2r_{\max} \epsilon_\pi}{1 - \gamma}.
\tag{25}
$$

*Proof.* Using the intermediate result of Lemma B.3, we have

$$
\begin{aligned}
&|\eta_1^{nm} - \eta_2^{nm}| \\
&\leq \left| \eta_1^{(n-1)m} - \eta_2^{(n-1)m} \right| + 2\gamma^{(n-1)m} \sum_{k=1}^{m} \gamma^k r_{\max} \left( (n-1)\delta^m + \delta^k + \epsilon_\pi \right) \\
&\leq \left| \eta_1^{(n-1)m} - \eta_2^{(n-1)m} \right| + 2\gamma^{(n-1)m} \sum_{k=1}^{m} \gamma^k r_{\max} \left( (n-1)m\epsilon_\pi + k\epsilon_\pi + \epsilon_\pi \right) \\
&\leq \left| \eta_1^0 - \eta_2^0 \right| + 2\epsilon_\pi \sum_{i=1}^{n} \gamma^{(i-1)m} \sum_{k=1}^{m} \gamma^k r_{\max} \left( (i-1)m + k + 1 \right) \\
&\leq 2\epsilon_\pi r_{\max} + 2\epsilon_\pi \sum_{i=1}^{n} \gamma^{(i-1)m} \sum_{k=1}^{m} \gamma^k r_{\max} \left( (i-1)m + k + 1 \right) \\
&= 2r_{\max} \left( \epsilon_\pi + \epsilon_\pi \sum_{i=1}^{n} \sum_{k=1}^{m} \gamma^k ((i-1)m + k) \gamma^{(i-1)m} + \epsilon_\pi \sum_{t=1}^{nm} \gamma^t \right) \\
&= 2r_{\max} \left( \epsilon_\pi + \epsilon_\pi \sum_{i=1}^{n} \sum_{k=1}^{m} ((i-1)m + k) \gamma^{(i-1)m+k} + \epsilon_\pi \sum_{t=1}^{nm} \gamma^t \right) \\
&= 2r_{\max} \left( \epsilon_\pi + \epsilon_\pi \sum_{t=1}^{nm} t\gamma^t + \epsilon_\pi \sum_{t=1}^{nm} \gamma^t \right).
\end{aligned}
\tag{26}
$$

Finally, while $n$ tends to be $\infty$, we have

$$
\begin{aligned}
|\eta_1 - \eta_2| &= \lim_{n \to \infty} |\eta_1^{nm} - \eta_2^{nm}| \\
&\leq 2r_{\max} \left( \epsilon_\pi + \epsilon_\pi \sum_{t=1}^{\infty} t\gamma^t + \epsilon_\pi \sum_{t=1}^{\infty} \gamma^t \right) = \frac{2\gamma r_{\max} \epsilon_\pi}{(1 - \gamma)^2} + \frac{2r_{\max} \epsilon_\pi}{1 - \gamma}.
\end{aligned}
\tag{27}
$$

$\square$

## C   THEORETICAL ANALYSES

**Theorem C.1** (Performance Bound). *Suppose the divergence of the $k$-step transition is bounded as*

$$\max_n \mathbb{E}_{s_{nm} \sim \rho^{\pi_D}} D_{\mathrm{TV}}(\mathbb{P}(s_{nm+k}|s_{nm})\|\hat{\mathbb{P}}(s_{nm+k}|s_{nm})) \leq \delta^k$$

*for each $k \in \{1, 2, \cdots, m\}$, and $\delta_{\max} = \max\limits_{k \in \{1,2,\cdots,m\}} \delta^k$. $\hat{\mathbb{P}}$ indicates the multi-step transition distribution induced by $\hat{T}$ and the policy $\pi$, and $\mathbb{P}$ indicates the multi-step transition distribution induced by the true dynamics $T$ and the policy $\pi$. The policy divergence is bounded as $\max_s D_{\mathrm{TV}}(\pi_D(a|s)\|\pi(a|s)) \leq \epsilon_\pi$. Then the return gap is bounded as*

$$|\eta(\pi) - \hat{\eta}(\pi)| \leq \frac{2\gamma r_{\max}\delta_{\max}}{(1-\gamma)(1-\gamma^m)} + \frac{2\gamma r_{\max}\epsilon_\pi}{(1-\gamma)^2} + \frac{4r_{\max}\epsilon_\pi}{1-\gamma}. \tag{28}$$

*Proof.*

$$\begin{aligned}
&|\eta(\pi) - \hat{\eta}(\pi)| \\
=&|\eta(\pi) - \eta(\pi_D) + \eta(\pi_D) - \hat{\eta}(\pi)| \\
=&\underbrace{|\eta(\pi) - \eta(\pi_D)|}_{L_1} + \underbrace{|\eta(\pi_D) - \hat{\eta}(\pi)|}_{L_2}
\end{aligned} \tag{29}$$

For $L_1$, we apply Lemma B.4 by using $\delta_{\max} \leq m\epsilon_\pi$ (no model error since $\eta(\pi)$ and $\eta(\pi_D)$ are both under the true dynamics) and obtain

$$L_1 \leq \frac{2\gamma r_{\max}\epsilon_\pi}{(1-\gamma)^2} + \frac{2r_{\max}\epsilon_\pi}{1-\gamma}. \tag{30}$$

For $L_2$, we directly use the result of Lemma B.3, that is

$$L_2 \leq \frac{2\gamma r_{\max}\delta_{\max}}{(1-\gamma)(1-\gamma^m)} + \frac{2r_{\max}\epsilon_\pi}{1-\gamma}. \tag{31}$$

Adding the bounds of $L_1$ and $L_2$ together yields the result. $\qquad\square$

**Assumption C.2** (Admissible Error Estimator). Assume that there exists a positive $b \in \mathbb{R}^+$ such that the following inequality equation 32 holds for any maximum backtracking length $m$ and any $s_t \in \mathcal{S}$, $a_t \in \mathcal{A}$.

$$D_{\mathrm{TV}}(\bar{T}_{\theta,m}(\cdot|s_t, a_t), T(\cdot|s_t, a_t)) \leq b \cdot \mathcal{U}^{\mathrm{ADM2}}(s_t, a_t), \tag{32}$$

where $\bar{T}_{\theta,m}$ is the overall conditioned distribution coming from

$$\bar{T}_{\theta,m}(\cdot|s_t, a_t) = \frac{1}{m} \sum_{k=1}^{m} \left[ \sum_{\substack{s_{t-k+1} \\ a_{t-k+1:t-1}}} \Gamma_\pi^{k-1}(s_{t-k+1}, a_{t-k+1:t-1}|s_t) \hat{T}_\theta(\cdot|s_{t-k+1}, a_{t-k+1:t}) \right], \tag{33}$$

and $\Gamma_\pi^k(s_{t-k+1:t}, a_{t-k+1:t}|s_{t+1})$ denotes the distribution over $(s_{t-k+1:t}, a_{t-k+1:t})$ conditioned on $s_{t+1}$ induced by the dynamic function $T$ and the policy $\pi$.

**Theorem C.3.** $\beta \cdot \mathcal{U}^{\mathrm{ADM2}}$ *is a valid $\xi$-uncertainty quantifier, with $\beta = b\frac{\gamma r_{\max}}{1-\gamma}$. Specifically,*

$$\left|\hat{\mathcal{T}}^\pi Q(s_t, a_t) - \mathcal{T}^\pi Q(s_t, a_t)\right| \leq \beta \cdot \mathcal{U}^{\mathrm{ADM2}}(s_t, a_t), \tag{34}$$

*where $\hat{\mathcal{T}}^\pi$ is the proxy operator induced by ADM-v2 to estimate the true Bellman operator $\mathcal{T}^\pi$.*

*Proof.* First, we define $y(\hat{s}_{t+1}, \hat{r}_{t+1}) = \hat{r}_{t+1} + \gamma \mathbb{E}_{a \sim \pi(\cdot|\hat{s}_{t+1})} [Q(\hat{s}_{t+1}, a)]$ and expand these two Bellman operator to

$$
\begin{aligned}
&\hat{\mathcal{T}}^\pi Q(s_t, a_t) \\
=&\mathbb{E}_{(s_{t-m+1:t-1}, a_{t-m+1:t-1}) \sim \Gamma_\pi^{m-1}(\cdot|s_t)} \left[ \frac{1}{m} \sum_{k=1}^m \mathbb{E}_{(\hat{s}_{t+1}, \hat{r}_{t+1}) \sim \hat{T}_\theta(\cdot|s_{t-k+1}, a_{t-k+1:t})} [y(\hat{s}_{t+1}, \hat{r}_{t+1})] \right] \\
=& \sum_{\substack{s_{t-m+1} \\ a_{t-m+1:t-1}}} \Gamma_\pi^{m-1}(s_{t-m+1}, a_{t-m+1:t-1}|s_t) \left[ \frac{1}{m} \sum_{k=1}^m \sum_{\substack{\hat{s}_{t+1} \\ \hat{r}_{t+1}}} \hat{T}_\theta(\cdot|s_{t-k+1}, a_{t-k+1:t}) y(\hat{s}_{t+1}, \hat{r}_{t+1}) \right] \\
=& \frac{1}{m} \sum_{k=1}^m \left[ \sum_{\substack{s_{t-k+1} \\ a_{t-k+1:t-1}}} \Gamma_\pi^{k-1}(s_{t-k+1}, a_{t-k+1:t-1}|s_t) \sum_{\substack{\hat{s}_{t+1} \\ \hat{r}_{t+1}}} \hat{T}_\theta(\cdot|s_{t-k+1}, a_{t-k+1:t})) y(\hat{s}_{t+1}, \hat{r}_{t+1}) \right] \\
=& \sum_{\hat{s}_{t+1}, \hat{r}_{t+1}} \bar{T}_{\theta,m}(\hat{s}_{t+1}, \hat{r}_{t+1}|s_t, a_t) y(\hat{s}_{t+1}, \hat{r}_{t+1}),
\end{aligned}
\tag{35}
$$

and

$$
\begin{aligned}
&\mathcal{T}^\pi Q(s_t, a_t) \\
=&\mathbb{E}_{\hat{s}_{t+1}, \hat{r}_{t+1} \sim T(\cdot|s_t, a_t)} [y(\hat{s}_{t+1}, \hat{r}_{t+1})] \\
=& \sum_{\hat{s}_{t+1}, \hat{r}_{t+1}} T(\hat{s}_{t+1}, \hat{r}_{t+1}|s_t, a_t) y(\hat{s}_{t+1}, \hat{r}_{t+1}).
\end{aligned}
\tag{36}
$$

Then, we can obtain

$$
\begin{aligned}
&\left| \hat{\mathcal{T}}^\pi Q(s_t, a_t) - \mathcal{T}^\pi Q(s_t, a_t) \right| \\
=& \sum_{\hat{s}_{t+1}, \hat{r}_{t+1}} \left| \bar{T}_{\theta,m}(\hat{s}_{t+1}, \hat{r}_{t+1}|s_t, a_t) - T(\hat{s}_{t+1}, \hat{r}_{t+1}|s_t, a_t) \right| \cdot |y(\hat{s}_{t+1}, \hat{r}_{t+1})| \\
=& \gamma \sum_{\hat{s}_{t+1}, \hat{r}_{t+1}} \left| \bar{T}_{\theta,m}(\hat{s}_{t+1}, \hat{r}_{t+1}|s_t, a_t) - T(\hat{s}_{t+1}, \hat{r}_{t+1}|s_t, a_t) \right| \cdot \left| \mathbb{E}_{a \sim \pi(\cdot|\hat{s}_{t+1})} [Q(\hat{s}_{t+1}, a)] \right| \\
\le& \frac{\gamma r_{\max}}{1-\gamma} \sum_{\hat{s}_{t+1}, \hat{r}_{t+1}} \left| \bar{T}_{\theta,m}(\hat{s}_{t+1}, \hat{r}_{t+1}|s_t, a_t) - T(\hat{s}_{t+1}, \hat{r}_{t+1}|s_t, a_t) \right| \\
=& \frac{\gamma r_{\max}}{1-\gamma} D_{\mathrm{TV}}(\bar{T}_{\theta,m}(\cdot|s_t, a_t), T(\cdot|s_t, a_t)) \\
\le& b \frac{\gamma r_{\max}}{1-\gamma} \mathcal{U}^{\mathrm{ADM2}}(s_t, a_t).
\end{aligned}
\tag{37}
$$

Thus, let $\beta = b\frac{\gamma r_{\max}}{1-\gamma}$, we can say that $\beta \cdot \mathcal{U}^{\mathrm{ADM2}}$ is a valid $\xi$-uncertainty quantifier. $\qquad\square$

## D  DETAILS OF IMPLEMENTATION

### D.1  ADM-V2 LEARNING

We present the pseudo code for learning ADM-v2 in Algorithm 1.

---

**Algorithm 1** ADM2-learn$(\hat{T}_\theta, \mathcal{D}_{\text{env}}, m, E)$

---

**Input**: Initial $\hat{T}_\theta$ with parameters $\theta$, dataset $\mathcal{D}_{\text{env}}$, maximum stride $m$, learning rate $\omega$, iterations $E$

1: **for** $E$ iterations **do**
2:     Randomly sample an integer $k$ from $[1, m]$ uniformly
3:     Sample $(s_t, a_{t:t+k-1}, r_{t+k}, s_{t+k})$ from $\mathcal{D}_{\text{env}}$
4:     Update ADM-v2 by $\theta \leftarrow \theta + \omega \nabla_\theta \log \hat{T}_\theta(s_{t+k}, r_{t+k}|s_t, a_{t:t+k-1})$
5: **end for**
6: **return** $\hat{T}_\theta$

---

### D.2  PARALLEL ANY-STEP ROLL-OUT

We present the pseudo code of PARoll (**P**arallel **A**ny-step **Roll**-out) in Algorithm 2. PARoll starts a roll-out from a given sequence $(s_0, a_0, s_1, a_1, \cdots, s_{m-2}, a_{m-2}, s_{m-1})$ sampled from the dataset $\mathcal{D}_{\text{env}}$. At each step, PARoll predicts the diverse states and rewards and estimates the any-step uncertainty in parallel. The parallel computation saves time costs compared to Lin et al. (2025).

---

**Algorithm 2** PARoll$(\hat{T}_\theta, \pi_\phi, H, (s_0, a_0, s_1, a_1, \cdots, s_{m-2}, a_{m-2}, s_{m-1}))$

---

**Input**: Learned ADM-v2 $\hat{T}_\theta$ with parameters $\theta$, policy $\pi_\phi$ with parameters $\phi$, maximum episode horizon $H$, initial state-action sequence $(s_0, a_0, s_1, a_1, \cdots, s_{m-2}, a_{m-2}, s_{m-1})$

1:   $\triangleright$ Roll-out Initialization
2: **for** $i = 1$ to $m$ **do**
3:     $h_{i-1}^{(i)} = \text{enc}_\theta(s_{i-1})$
4:     **for** $t = i - 1$ to $m - 2$ **do**
5:         $h_{t+1}^{(i)} = g_\theta(h_t^{(i)}, a_t)$
6:     **end for**
7: **end for**
8:
9: **for** $t = m - 1$ to $H - 1$ **do**
10:     Choose action by $a_t \sim \pi(\cdot|\hat{s}_t)$ ($a_t \sim \pi(\cdot|s_t)$ if $t = m - 1$)
11:     Update hidden states in parallel by $[h_{t+1}^{(i)}]_{i=1}^m = [g_\theta(h_t^{(i)}, a_t)]_{i=1}^m$
12:     Predict diverse states and rewards in parallel by
    $[(\mu_{s_{t+1}}^{(i)}, \Sigma_{s_{t+1}}^{(i)}, \mu_{r_{t+1}}^{(i)}, \Sigma_{r_{t+1}}^{(i)})]_{i=1}^m = [\text{dec}_\theta(h_{t+1}^{(i)})]_{i=1}^m,$
    $[\hat{s}_{t+1}^{(i)}]_{i=1}^m \sim [\mathcal{N}(\mu_{s_{t+1}}^{(i)}, \Sigma_{s_{t+1}}^{(i)})]_{i=1}^m, [\hat{r}_{t+1}^{(i)}]_{i=1}^m \sim [\mathcal{N}(\mu_{r_{t+1}}^{(i)}, \Sigma_{r_{t+1}}^{(i)})]_{i=1}^m$
13:     Uniformly choose the state and reward prediction by
    $\hat{s}_{t+1} \sim \text{Uniform}\{\hat{s}_{t+1}^{(1)}, \hat{s}_{t+1}^{(2)}, \cdots, \hat{s}_{t+1}^{(m)}\}$ and $\hat{r}_{t+1} \sim \text{Uniform}\{\hat{r}_{t+1}^{(1)}, \hat{r}_{t+1}^{(2)}, \cdots, \hat{r}_{t+1}^{(m)}\}$
14:     Estimate the model uncertainty by $u_t = \mathcal{U}^{\text{ADM2}}(\hat{s}_t, a_t)$
15:     Reset the hidden state at the stride edge $h_{t+1}^{((t \bmod m)+2)} = \text{enc}_\theta(\hat{s}_{t+1}^{((t \bmod m)+2)})$
16: **end for**
17: **return** $(s_{m-1}, a_{m-1}, u_{m-1}, \hat{r}_m, \hat{s}_m, \cdots, \hat{s}_{H-1}, a_{H-1}, u_{H-1}, \hat{r}_H, \hat{s}_H)$

---

### D.3  ADM-V2 FOR OFF-POLICY EVALUATION

When the dynamics model has sufficient predictive capability, a straightforward off-policy evaluation (OPE) method is to treat the dynamics model as a data-driven simulator and allow the policy to roll out within it. We let the policy roll out $N$ full-horizon trajectories in ADM-v2, and then compute the average return of these trajectories as the result of OPE. The pseudo code is shown in Algorithm 3.

---

**Algorithm 3** ADM2OPE($\hat{T}_\theta, \pi_\phi, \mathcal{D}_{\text{env}}, H, N$)

---

**Input**: Learned ADM-v2 $\hat{T}_\theta$ with parameters $\theta$, policy $\pi_\phi$ with parameters $\phi$, offline dataset $\mathcal{D}_{\text{env}}$, maximum episode horizon $H$, number of episodes $N$

1:  **for** $j = 1$ to $N$ **do**
2:      Sample a sequence $(s_0, a_0, r_1, s_1, a_1, r_2, \cdots, s_{m-2}, a_{m-2}, r_{m-1}, s_{m-1})$ from $\mathcal{D}_{\text{env}}$
3:      Model roll-out by PARoll($\hat{T}_\theta, \pi_\phi, H, (s_0, a_0, s_1, a_1, \cdots, s_{m-2}, a_{m-2}, s_{m-1})$)
4:      Calculate the return by $\hat{R}_j = \sum_{\tau=m}^{H} \gamma^{\tau-m} \hat{r}_\tau$
5:  **end for**
6:  **return** $\hat{R} = \frac{1}{N} \sum_{j=1}^{N} \hat{R}_j$

---

## D.4 ADM-v2 for Offline Policy Optimization

Roll-out data in ADM-v2 can be used to optimize the policy offline. We follow the framework of MOPO (Yu et al., 2020), but replace the ensemble dynamics model in MOPO with ADM-v2 and set the roll-out length as the maximum episode horizon $H$ of the task. The pseudo code of ADM-v2 for offline policy optimization is shown in Algorithm 4.

---

**Algorithm 4** ADM2PO-fh($\hat{T}_\theta, \pi_\phi, \mathcal{D}_{\text{env}}, H, E, \beta$)

---

**Input**: Learned ADM-v2 $\hat{T}_\theta$ with parameters $\theta$, policy $\pi_\phi$ with parameters $\phi$, offline dataset $\mathcal{D}_{\text{env}}$, maximum episode horizon $H$, model data buffer $\mathcal{D}_{\text{model}}$, iterations $E$, penalty coefficient $\beta$

1:  **for** $E$ iterations **do**
2:      **for** $M$ model roll-outs **do**
3:          Sample a sequence $(s_0, a_0, r_1, s_1, a_1, r_2, \cdots, s_{m-2}, a_{m-2}, r_{m-1}, s_{m-1})$ from $\mathcal{D}_{\text{env}}$
4:          Model roll-out by PARoll($\hat{T}_\theta, \pi_\phi, H, (s_0, a_0, s_1, a_1, \cdots, s_{m-2}, a_{m-2}, s_{m-1})$)
5:          Penalize the reward by $\tilde{r}_t = r_t - \beta \mathcal{U}^{\text{ADM2}}(s_t, a_t)$ for each roll-out step $t$
6:          Add the penalized model roll-out data to $\mathcal{D}_{\text{model}}$
7:      **end for**
8:  **end for**
9:  **for** $G$ policy updates **do**
10:     Update current policy $\pi_\phi$ by SAC (Haarnoja et al., 2018) using samples from $\mathcal{D}_{\text{env}} \cup \mathcal{D}_{\text{model}}$
11: **end for**
12: **return** $\pi_\phi$

---

The policy optimization method used in Algorithm 4 is SAC (Haarnoja et al., 2018), following MOPO (Yu et al., 2020) and ADMPO (Lin et al., 2025).

## D.5 Network Structure

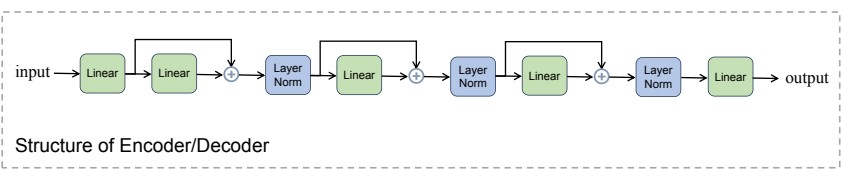

Figure 8: The detailed network structure of the encoder and the decoder.

Both the state encoder and the transition decoder of ADM-v2 adopt a 5-layer MLP architecture, where the middle three hidden layers are equipped with skip connections (He et al., 2016) and layer normalization (Ba et al., 2016), as depicted in Figure 8. The GRU in ADM-v2 does not incorporate any special modifications; it follows the same architecture as in Cho et al. (2014) and is implemented directly using `torch.nn.GRU`[1].

---

[1]https://pytorch.org/

# E    DETAILS OF DYNAMICS MODEL BASELINES

## E.1    ANY-STEP DYNAMICS MODEL

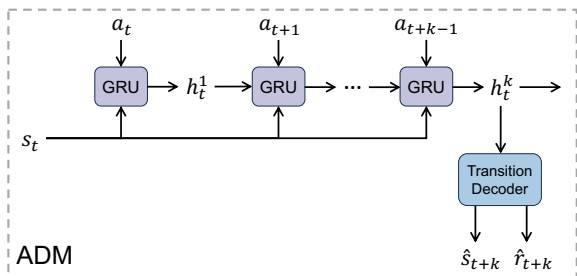

Figure 9: Illustration of the first version of Any-step Dynamics Model (ADM), which is structured using a GRU Cell.

Lin et al. (2025) propose the original version of Any-step Dynamics Model (ADM). Multi-step dynamics models (Asadi et al., 2018; 2019; Che et al., 2018) take $s_t$ along with a $k$-step sequence of actions $(a_t, a_{t+1}, \cdots, a_{t+k-1})$ as inputs to predict $s_{t+k}$ and $r_{t+k}$. ADM extends the multi-step dynamics model to allow $k$ to be any positive integer within a specified range, as delineated in Definition E.1.

**Definition E.1** (Any-step Dynamics Model). Given the maximum stride $m$, an any-step dynamics model $\hat{T}_{\mathrm{ADM}}(s_{t+k}, r_{t+k}|s_t, a_{t:t+k-1})$ is the distribution of $s_{t+k} \in \mathcal{S}$ and $r_{t+k} \in \mathbb{R}$ conditioned on $(s_t, a_{t:t+k-1}) = (s_t, a_t, a_{t+1}, \cdots, a_{t+k-1}) \in \mathcal{S} \times \mathcal{A}^k$, where $k$ can be any integer between $[1, m]$.

To handle inputs with variable step sizes, ADM utilizes an RNN (Elman, 1990) with a GRU (Cho et al., 2014) cell, as depicted in Figure 9. Since the input state consists of only one step, while the action sequence is of multiple steps, ADM duplicates the state to match the length of the action sequence, and then sequentially feeds it into the RNN. The input $(s_t, a_{t:t+k-1})$, after being represented by the RNN, yields the hidden $h_t^k$, which is then fed into an MLP to obtain the mean and standard deviation of $s_{t+k}$ and $r_{t+k}$. The transition decoder follows the same architecture as ADM-v2, as depicted in Figure 8.

---

**Algorithm 5** Roll-out in ADM: **ADM-Roll**($\hat{T}_{\mathrm{ADM}}, \pi_\phi, H, (s_0, a_0, s_1, a_1, \cdots, s_{m-2}, a_{m-2}, s_{m-1})$)

---

**Input**: Learned $\hat{T}_{\mathrm{ADM}}$, policy $\pi_\phi$ with parameters $\phi$, maximum episode horizon $H$, initial state-action sequence $(s_0, a_0, s_1, a_1, \cdots, s_{m-2}, a_{m-2}, s_{m-1})$

1: **for** $t = m - 1$ to $H - 1$ **do**
2:     **if** $t = m - 1$ **then** Sample $a_t \sim \pi_\phi(\cdot|s_t)$
3:     **else** Sample $a_t \sim \pi_\phi(\cdot|\hat{s}_t)$
4:     Randomly sample an integer $k$ from $[1, m]$ uniformly
5:     **if** $t \leq m + k - 2$ **then** Roll out via $(\hat{s}_{t+1}, \hat{r}_{t+1}) \sim \hat{T}_{\mathrm{ADM}}(\cdot|s_{t+1-k}, a_{t+1-k:t})$
6:     **else** Roll out via $(\hat{s}_{t+1}, \hat{r}_{t+1}) \sim \hat{T}_{\mathrm{ADM}}(\cdot|\hat{s}_{t+1-k}, a_{t+1-k:t})$
7:     Estimate the model uncertainty by $u_t = \mathcal{U}^{\mathrm{ADM}}(\hat{s}_t, a_t)$
8: **end for**
9: **return** $(s_{m-1}, a_{m-1}, u_{m-1}, \hat{r}_m, \hat{s}_m, \cdots, \hat{s}_{H-1}, a_{H-1}, u_{H-1}, \hat{r}_H, \hat{s}_H)$

---

With ADM, the frequent bootstrapping during model roll-out can be reduced. Specifically, given the maximum stride $m$, a state-action sequence of length $m$, $(s_1, a_1, s_2, a_2, \cdots, s_m, a_m)$, is sampled from the data buffer to start the roll-out. To obtain the prediction $\hat{s}_{m+1}$, an integer between $[1, m]$ is chosen uniformly at random as the backtracking length. If $k$ steps are selected for backtracking, $(s_{m-k+1}, a_{m-k+1:m})$ will be fed into $\hat{T}_{\mathrm{ADM}}$ to obtain the prediction result. After rolling out several steps, it inevitably traces back to previously predicted states. For example, for the prediction $\hat{s}_{2m}$, it can only backtrack to one of $(s_m, \hat{s}_{m+1}, \hat{s}_{m+2}, \cdots, \hat{s}_{2m-1})$. The backtracked state, one part of the attribution for the next state prediction, is located several steps ahead, in expectation. Thus, ADM

reduces the actual bootstrapping count of a rolled-out trajectory. The pseudo-code of roll-out in ADM is shown in Algorithm 5.

The difference among probabilistic predictions $\hat{T}_{\mathrm{ADM}}(\cdot|s_{t-k+1}, a_{t-k+1:t})$ obtained with different backtracking $k$ serves as a natural measure of model uncertainty, which can be quantified using variance (or standard deviation), as defined by

$$\mathcal{U}^{\mathrm{ADM}}(s_t, a_t) = \mathbb{E}\left[\left\|\mathrm{Var}_{k\sim\mathrm{Uniform}(m), \hat{s}_{t+1}\sim\hat{T}_{\mathrm{ADM}}(\cdot|s_{t-k+1}, a_{t-k+1:t})}\left[\hat{s}_{t+1}\right]\right\|\right]. \qquad (38)$$

### E.2 Vanilla RNN Dynamics Model

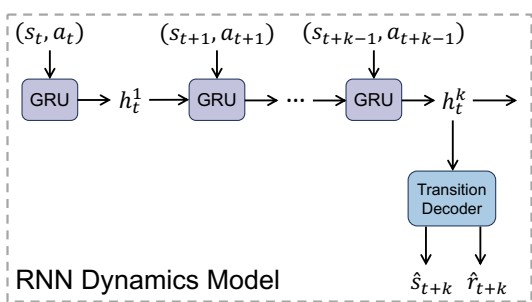

Figure 10: Illustration of the RNN dynamics model, which is structured using a GRU Cell.

Vanilla RNN dynamics model shares the same structure as ADM but makes the prediction $\hat{s}_{t+1}$ using the historical state-action sequence $(\hat{s}_{t-k+1}, a_{t-k+1}, \cdots, \hat{s}_t, a_t)^2$ as input, as depicted in Figure 10. The transition decoder follows the same architecture as ADM-v2, as shown in Figure 8. To match the paradigm of ADM and ADM-v2, $k$ is uniformly sampled from $[1, m]$. Since the predicted current state is one part of the input, the prediction error of the next state is accumulated based on the deviation of the current state. The bootstrapping is as frequent as the ensemble dynamics model (Janner et al., 2019; Lin et al., 2023; Yu et al., 2020).

The difference among probabilistic predictions $\hat{T}_{\mathrm{RNN}}(\cdot|s_{t-k+1:t}, a_{t-k+1:t})$ obtained with different $k$ also serves as a model uncertainty, which can be quantified using variance (or standard deviation), as defined by

$$\mathcal{U}^{\mathrm{RNN}}(s_t, a_t) = \mathbb{E}\left[\left\|\mathrm{Var}_{k\sim\mathrm{Uniform}(m), \hat{s}_{t+1}\sim\hat{T}_{\mathrm{RNN}}(\cdot|s_{t-k+1:t}, a_{t-k+1:t})}\left[\hat{s}_{t+1}\right]\right\|\right]. \qquad (39)$$

---

**Algorithm 6** Roll-out in RNN: **RNN-Roll**$(\hat{T}_{\mathrm{RNN}}, \pi_\phi, H, (s_0, a_0, s_1, a_1, \cdots, s_{m-2}, a_{m-2}, s_{m-1}))$

---

**Input**: Learned $\hat{T}_{\mathrm{RNN}}$, policy $\pi_\phi$ with parameters $\phi$, maximum episode horizon $H$, initial state-action sequence $(s_0, a_0, s_1, a_1, \cdots, s_{m-2}, a_{m-2}, s_{m-1})$

1: **for** $t = m - 1$ to $H - 1$ **do**
2:     **if** $t = m - 1$ **then** Sample $a_t \sim \pi_\phi(\cdot|s_t)$
3:     **else** Sample $a_t \sim \pi_\phi(\cdot|\hat{s}_t)$
4:     Randomly sample an integer $k$ from $[1, m]$ uniformly
5:     **if** $t \leq m + k - 2$ **then** Roll out via $(\hat{s}_{t+1}, \hat{r}_{t+1}) \sim \hat{T}_{\mathrm{RNN}}(\cdot|s_{t+1-k:t}, a_{t+1-k:t})$
6:     **else** Roll out via $(\hat{s}_{t+1}, \hat{r}_{t+1}) \sim \hat{T}_{\mathrm{RNN}}(\cdot|\hat{s}_{t+1-k:t}, a_{t+1-k:t})$
7:     Estimate the model uncertainty by $u_t = \mathcal{U}^{\mathrm{RNN}}(\hat{s}_t, a_t)$
8: **end for**
9: **return** $(s_{m-1}, a_{m-1}, u_{m-1}, \hat{r}_m, \hat{s}_m, \cdots, \hat{s}_{H-1}, a_{H-1}, u_{H-1}, \hat{r}_H, \hat{s}_H)$

---

The pseudo-code of roll-out in RNN is shown in Algorithm 6.

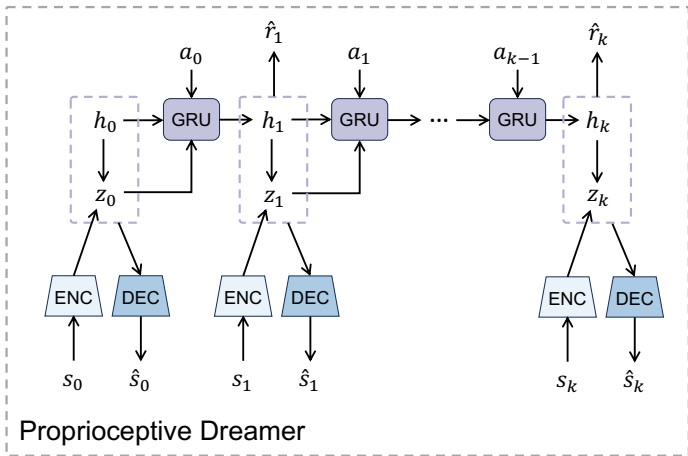

Figure 11: Illustration of the proprioceptive Dreamer, which is structured using a GRU Cell.

### E.3 PROPRIOCEPTIVE DREAMER

Dreamer series (Hafner et al., 2020; 2021; 2023) focuses on the environment modeling in visual control tasks where the observation is of the image form. We modify the architecture of Dreamer to adapt to the proprioceptive control tasks where the observation is just a vector, as depicted in Figure 11. Inner modules like GRU, encoder, and decoder share the same network structure as ADM-v2 and other baselines.

The world model in Dreamer consists of a recurrent state-space model that maintains a latent representation of the environment. It has a prior that predicts the next latent state from the current state and action, and a posterior that refines this prediction using the actual observation. A decoder reconstructs observations or rewards from the latent state, while a recurrent core integrates information across time. This latent dynamics model enables the agent to imagine trajectories for policy learning.

---

**Algorithm 7 Dreamer-Roll**($\hat{T}_{\text{Dreamer}}, \pi_\phi, H, (s_0, a_0, s_1, a_1, \cdots, s_{m-2}, a_{m-2}, s_{m-1})$)

---

**Input**: Learned $\hat{T}_{\text{Dreamer}}$, policy $\pi_\phi$ with parameters $\phi$, maximum episode horizon $H$, initial state-action sequence $(s_0, a_0, s_1, a_1, \cdots, s_{m-2}, a_{m-2}, s_{m-1})$

1: **for** $\tau = 0$ to $m - 1$ **do**
2:     Encode $s_\tau$ to $z_\tau$
3: **end for**
4: **for** $t = m - 1$ to $H - 1$ **do**
5:     **if** $t = m - 1$ **then** Sample $a_t \sim \pi_\phi(\cdot|s_t)$
6:     **else** Sample $a_t \sim \pi_\phi(\cdot|\hat{s}_t)$
7:     Randomly sample an integer $k$ from $[1, m]$ uniformly
8:     **if** $t \le m + k - 2$ **then** Roll out via $(\hat{z}_{t+1}, \hat{r}_{t+1}) \sim \hat{T}_{\text{Dreamer}}(\cdot|z_{t+1-k:t}, a_{t+1-k:t})$
9:     **else** Roll out via $(\hat{z}_{t+1}, \hat{r}_{t+1}) \sim \hat{T}_{\text{Dreamer}}(\cdot|\hat{z}_{t+1-k:t}, a_{t+1-k:t})$
10:     Decode $\hat{z}_{t+1}$ to $\hat{s}_{t+1}$
11:     Estimate the model uncertainty by $u_t = \mathcal{U}^{\text{Dreamer}}(\hat{z}_t, a_t)$
12: **end for**
13: **return** $(s_{m-1}, a_{m-1}, u_{m-1}, \hat{r}_m, \hat{s}_m, \cdots, \hat{s}_{H-1}, a_{H-1}, u_{H-1}, \hat{r}_H, \hat{s}_H)$

---

The pseudo-code of roll-out in proprioceptive Dreamer is shown in Algorithm 7.

The difference among probabilistic predictions $\hat{T}_{\text{Dreamer}}(\cdot|z_{t-k+1:t}, a_{t-k+1:t})$ obtained with different $k$ also serves as a model uncertainty, which can be quantified using variance (or standard deviation),

---

[2]It would be $(s_{t-k+1}, a_{t-k+1}, \cdots, s_t, a_t)$ from the dataset if it were at the beginning of the roll-out.

as defined by

$$\mathcal{U}^{\text{Dreamer}}(z_t, a_t) = \mathbb{E}\left[\left\|\text{Var}_{k\sim\text{Uniform}(m),\hat{z}_{t+1}\sim\hat{T}_{\text{Dreamer}}(\cdot|z_{t-k+1:t},a_{t-k+1:t})}\left[\hat{z}_{t+1}\right]\right\|\right]. \quad (40)$$

This uncertainty estimator for Dreamer lies in the latent space.

### E.4 ENSEMBLE DYNAMICS MODEL

The ensemble dynamics model (EDM) is applied in many previous model-based methods (Janner et al., 2019; Lin et al., 2023; Yu et al., 2020; Kidambi et al., 2020; Sun et al., 2023; Luo et al., 2024). Given the ensemble size $N_{\text{en}}$, $N_{\text{en}}$ MLP-based single-step dynamics models are learned by maximizing the likelihood, $\{\hat{T}_i(s_{t+1}, r_{t+1}|s_t, a_t)\}_{i=1}^{N_{\text{en}}}$. The difference among probabilistic predictions obtained from different learners serves as an uncertainty estimation, which can be quantified using variance (or standard deviation), as defined by

$$\mathcal{U}^{\text{EN}}(s_t, a_t) = \mathbb{E}\left[\left\|\text{Var}_{k\sim\text{Uniform}(N_{\text{EN}}),\hat{s}_{t+1}\sim\hat{T}_k(\cdot|s_t,a_t)}\left[\hat{s}_{t+1}\right]\right\|\right]. \quad (41)$$

---

**Algorithm 8** Roll-out in EDM: **EDM-Roll**($\{\hat{T}_i\}_{i=1}^{N_{\text{EN}}}, \pi_\phi, H, s_0$)

---

**Input**: Learned $\{\hat{T}_i\}_{i=1}^{N_{\text{EN}}}$, policy $\pi_\phi$ with parameters $\phi$, maximum episode horizon $H$, initial state $s_0$
1: **for** $t = 0$ to $H - 1$ **do**
2:     **if** $t = 0$ **then** Sample $a_t \sim \pi_\phi(\cdot|s_t)$
3:     **else** Sample $a_t \sim \pi_\phi(\cdot|\hat{s}_t)$
4:     Randomly sample an integer $k$ from $[1, N_{\text{EN}}]$ uniformly
5:     **if** $t = 0$ **then** Roll out via $(\hat{s}_{t+1}, \hat{r}_{t+1}) \sim \hat{T}_k(\cdot|s_t, a_t)$
6:     **else** Roll out via $(\hat{s}_{t+1}, \hat{r}_{t+1}) \sim \hat{T}_k(\cdot|\hat{s}_t, a_t)$
7:     Estimate the model uncertainty by $u_t = \mathcal{U}^{\text{EN}}(\hat{s}_t, a_t)$
8: **end for**
9: **return** $(s_0, a_0, u_0, \hat{r}_1, \hat{s}_1, \cdots, \hat{s}_{H-1}, a_{H-1}, u_{H-1}, \hat{r}_H, \hat{s}_H)$

---

The pseudo-code of roll-out in EDM is shown in Algorithm 8.

# F    EXPERIMENTAL DETAILS

## F.1    SETTINGS OF OFFLINE POLICY OPTIMIZATION AND EVALUATION

We reveal the experimental settings of ADM-v2 for off-policy evaluation in Section 4.2 and offline policy optimization in ADM-v2 in Section 4.3 below. Table 3 lists three important hyper-parameters, the maximum stride $m$, the roll-out horizon $H$, and the penalty coefficient, under different tasks.

Table 3: Hyper-parameter settings of ADM-v2 under different tasks.

| Domain Name | Task Name | $m$ | $H$ | $\beta$ |
|---|---|---|---|---|
| D4RL MuJoCo | hopper-random | 5 | 1000 | 100 |
| | halfcheetah-random | 3 | 1000 | 0.05 |
| | walker2d-random | 5 | 1000 | 5 |
| | hopper-medium | 6 | 1000 | 20 |
| | halfcheetah-medium | 2 | 1000 | 3 |
| | walker2d-medium | 5 | 1000 | 2.5 |
| | hopper-medium-replay | 5 | 1000 | 5 |
| | halfcheetah-medium-replay | 2 | 1000 | 5 |
| | walker2d-medium-replay | 5 | 1000 | 0.25 |
| | hopper-medium-expert | 2 | 1000 | 100 |
| | halfcheetah-medium-expert | 2 | 1000 | 3 |
| | walker2d-medium-expert | 5 | 1000 | 2.5 |
| NeoRL MuJoCo | neorl-hopper-L | 5 | 1000 | 25 |
| | neorl-halfcheetah-L | 2 | 1000 | 2 |
| | neorl-walker2d-L | 5 | 1000 | 3 |
| | neorl-hopper-M | 5 | 1000 | 20 |
| | neorl-halfcheetah-M | 2 | 1000 | 2 |
| | neorl-Walker2d-M | 5 | 1000 | 3 |
| | neorl-hopper-H | 4 | 1000 | 25 |
| | neorl-halfcheetah-H | 2 | 1000 | 5 |
| | neorl-walker2d-H | 5 | 1000 | 3 |

We set the roll-out length $H$ as 1000, which is the maximum episodic length of MuJoCo (Todorov et al., 2012) tasks.

Table 4 lists other hyper-parameters for learning ADM-v2, kept unchanging under different tasks.

Table 4: Other hyper-parameters for learning ADM-v2.

| Hyper-parameter | Value | Description |
|---|---|---|
| $L_{\mathrm{enc}}$ | 5 | number of layers in state encoder. |
| $h_{\mathrm{enc}}$ | 200 | size of each hidden layer in state encoder. |
| $L_{\mathrm{dec}}$ | 5 | number of leyers in transition decoder. |
| $h_{\mathrm{dec}}$ | 200 | size of each hidden layer in transition decoder. |
| $L_{\mathrm{gru}}$ | 3 | number of GRU layers. |
| $h_{\mathrm{gru}}$ | 200 | size of the hidden state in each GRU layer. |
| $lr_{\mathrm{adm2}}$ | $3 \times 10^{-4}$ | learning rate of ADM-v2. |
| optimizer | Adam | optimizer of ADM-v2. |
| $E_{\mathrm{adm2}}$ | 100 | number of epochs for learning ADM-v2. |
| $B_{\mathrm{learn}}$ | 1024 | batch size during model learning. |
| $B_{\mathrm{roll}}$ | 4096 | batch size during model roll-out. |
| seeds | (0, 1, 2, 3, 4) | seeds for experiments. |

The policy optimization method used in ADM2PO-fh is SAC (Haarnoja et al., 2018). We follow the standard implementation of SAC, with hyper-parameters listed in Table 5.

Table 5: Hyper-parameters of offline policy optimization in ADM-v2.

| Hyper-parameter | Value | Description |
|---|---|---|
| $N_Q$ | 2 | the number of critics. |
| actor network | FC(256,256) | fully connected (FC) layers with ReLU activations. |
| critic network | FC(256,256) | fully connected (FC) layers with ReLU activations. |
| $\tau$ | $5 \times 10^{-3}$ | target network smoothing coefficient. |
| $\gamma$ | 0.99 | discount factor. |
| $lr_{\text{actor}}$ | $1 \times 10^{-4}$ | learning rate of actor. |
| $lr_{\text{critic}}$ | $3 \times 10^{-4}$ | learning rate of critic. |
| optimizer | Adam | optimizers of the actor and critics. |
| batch size | 256 | batch size for each update. |
| $E_{\text{sac}}$ | $1 \times 10^{6}$ | number of steps for policy optimization. |
| seeds | (0, 1, 2, 3, 4) | seeds for experiments. |

## F.2 SOURCE OF BASELINES' RESULTS

For the evaluation on D4RL (Fu et al., 2020) benchmarks, the results of the compared baselines come from three sources:

- Retraining on D4RL datasets of v2 version with OfflineRL-Kit (Sun, 2023), for the algorithms whose original papers only report the performance on the v0 version, such as CQL (Kumar et al., 2020), MOPO (Yu et al., 2020).

- Including the scores in their papers, for the algorithms whose original papers report the performance on the v2 version, such as EDAC (An et al., 2021), MOBILE (Sun et al., 2023), MOREC (Luo et al., 2024), and ADMPO (Lin et al., 2025).

- Implementing in our code, for policy optimization with full-horizon roll-out in dynamics model baselines, including EDM, RDM, P-Dreamer, and ADM.

For the evaluation on NeoRL (Qin et al., 2022) benchmarks, we report the scores of CQL and MOPO from the original paper of NeoRL, retrain EDAC with OfflineRL-Kit (Sun, 2023), include the scores in papers of MOBILE, MOREC, and ADMPO, and implement the code of policy optimization with full-horizon roll-out in EDM, RDM, P-Dreamer, and ADM.

# G ADDITIONAL EMPIRICAL RESULTS

## G.1 DIRECT PREDICTION VERSUS BOOTSTRAPPING PREDICTION

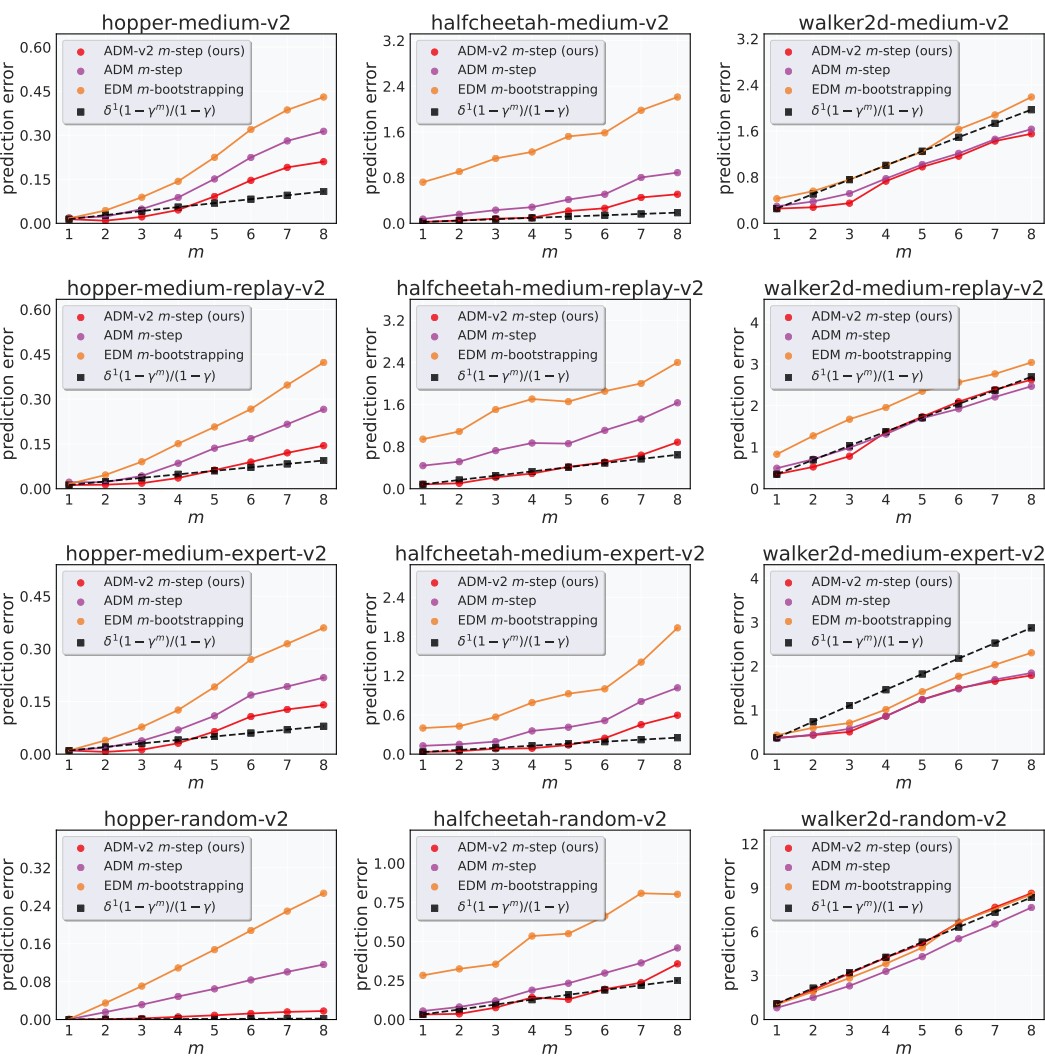

Figure 12: Comparison between direct prediction and bootstrapping prediction. We plot the $m$-step direct prediction error curve of ADM-v2 and ADM, the $m$-step bootstrapping prediction error curve of EDM, and the curve of $\delta^1 \frac{1-\gamma^m}{1-\gamma}$, with $m$ varying from 1 to 8. Results are averaged over 5 seeds.

We compare the direct prediction error and the bootstrapping prediction error, as demonstrated in Figure 12. The experiment is conducted on twelve D4RL (Fu et al., 2020) MuJoCo datasets. For each dataset, we divide the data into two parts, one for training and another for evaluation. We sample 100 sequences from the evaluation dataset and calculate the average error for each $m$. We can find a proper $m$, with which $\delta_{\max} = \max_{k \in \{1, 2, \cdots, m\}} \delta^k < \frac{\delta^1 (1 - \gamma^m)}{1 - \gamma}$ is satisfied, from the evaluation results of almost all datasets. While setting $m$ as the proper value, the performance bound of ADM-v2 is tighter than that of single-step dynamics models such as EDM. Furthermore, the direct prediction error of ADM-v2 is smaller than that of ADM and the bootstrapping prediction error of EDM in most cases.

## G.2 ROLL-OUT ERROR

We compare ADM-v2 with four dynamics models in terms of the growth curve of the compounding error (in log scale) as the roll-out horizon increases to the maximum episode length, as demonstrated in Figure 13. The experiment is conducted on twelve D4RL (Fu et al., 2020) MuJoCo datasets. For each dataset, we divide the data into two parts, one for training and another for evaluation. We sample 100 trajectories from the evaluation dataset and execute actions sequentially in each trained dynamics model to calculate the average roll-out error at each step. We observe that the roll-out error curve of ADM-v2 is below the curve of other baselines in most cases.

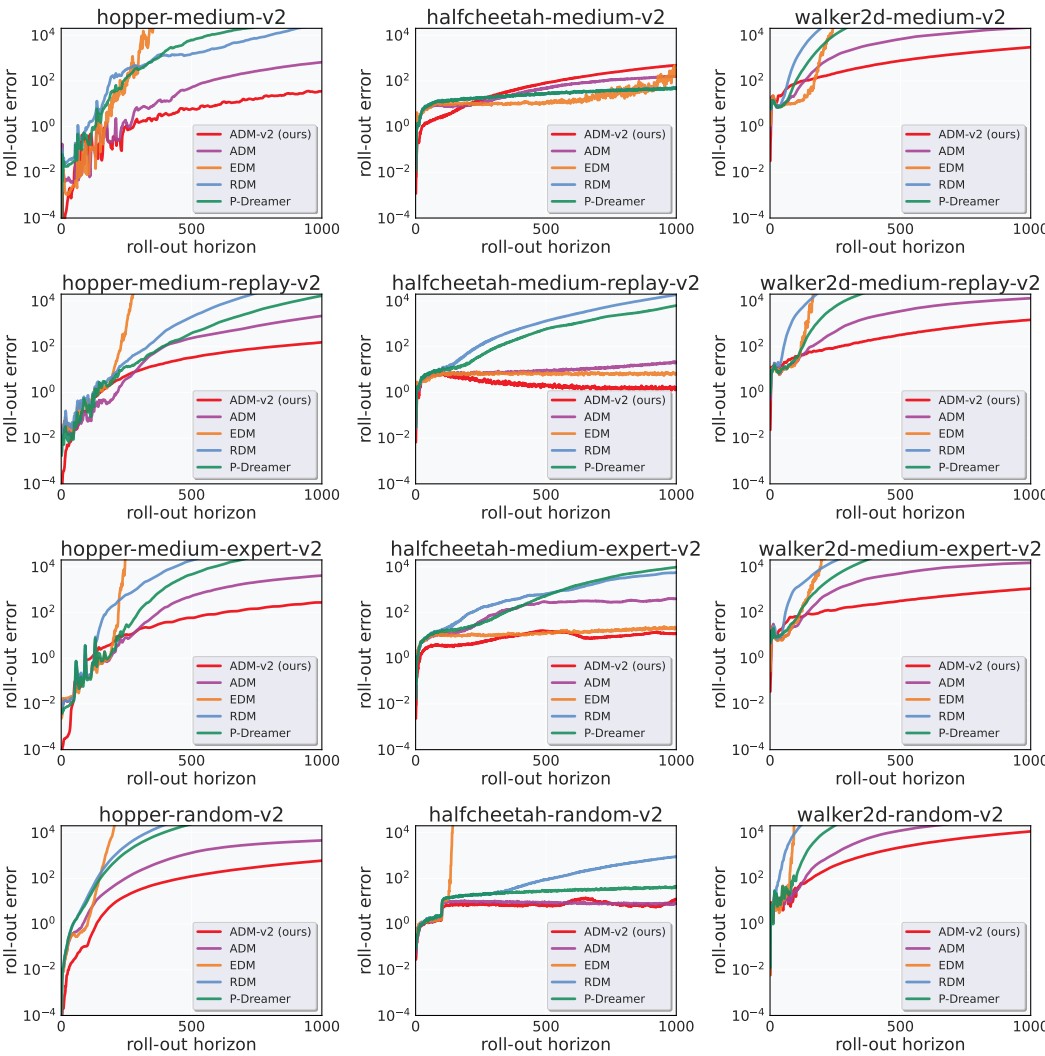

Figure 13: Growth curves of the compounding error (in log scale) as the roll-out horizon increases to the maximum episode length. We compare ADM-v2 with four baselines, including ADM, EDM, RDM, and P-Dreamer, on twelve D4RL MuJoCo datasets. Results are averaged over 5 seeds.

## G.3 ROLL-OUT EFFICIENCY

We compare ADM-v2 with four dynamics models in terms of roll-out throughput (the average number of roll-out samples per second). Since the state and action dimensions of different types of datasets in each MuJoCo (Todorov et al., 2012) task are the same, the corresponding dynamics models also have the same size. Therefore, we only conduct experiments on the medium-type datasets of three MuJoCo

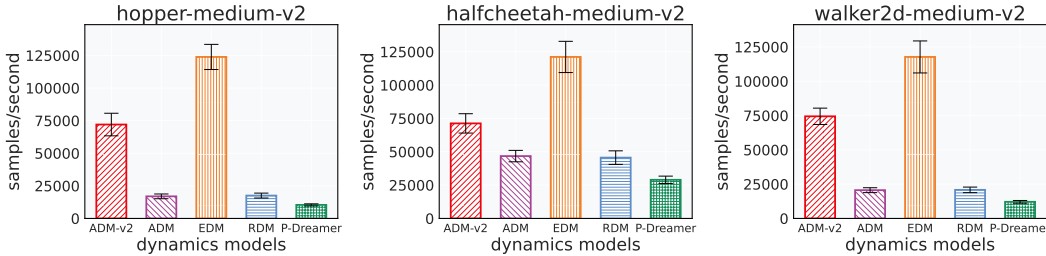

Figure 14: Comparison of ADM-v2 with ADM, EDM, RDM, and P-Dreamer in terms of roll-out throughput on three D4RL MuJoCo datasets. Results are averaged over 5 seeds.

tasks. The results are demonstrated in Figure 14. The roll-out throughput of ADM-v2 is greater than other RNN-based dynamics models, including ADM, RDM, and P-Dreamer. EDM achieves high time efficiency in roll-out due to its simple model architecture. However, the disadvantage of ADM-v2 compared to EDM is not significant. Considering the fidelity advantage of ADM-v2 in full-horizon roll-outs, this slight cost in efficiency is acceptable.

Table 6: Model capacity and GPU memory consumption during roll-out.

| Task Name | Model Capacity | | | | | GPU Memory Consumption | | | | |
|---|---|---|---|---|---|---|---|---|---|---|
| | ADM-v2 | ADM | EDM | RDM | P-Dreamer | ADM-v2 | ADM | EDM | RDM | P-Dreamer |
| hopper | 1.057M | 0.779M | 1.798M | 0.779M | 1.163M | 4.4G | 4.4G | 4.8G | 4.4G | 4.5G |
| halfcheetah | 1.063M | 0.787M | 1.857M | 0.787M | 1.169M | 4.7G | 4.6G | 5.1G | 4.6G | 4.8G |
| walker2d | 1.063M | 0.787M | 1.857M | 0.787M | 1.169M | 4.8G | 4.7G | 5.1G | 4.7G | 4.9G |

We list the model capacity and GPU memory consumption during roll-out for each dynamics model in Table 6. Due to architectural changes, ADM-v2 has a larger number of parameters than ADM, which indeed increases the computational cost of a single forward pass. However, by leveraging the PARoll algorithm designed for ADM-v2, the model removes state backtracking and significantly reduces the number of forward passes required during roll-out. As a result, ADM-v2 achieves higher roll-out throughput compared to ADM. In terms of GPU memory consumption, ADM-v2 does not require additional resources.

## G.4 OFF-POLICY EVALUATION

We conduct the Off-policy Evaluation (OPE) experiment on the DOPE (Fu et al., 2021) benchmark. The DOPE benchmark provides some policies for each task and proposes several metrics to measure the consistency between the estimated returns by the OPE method and the true returns. We denote $N$ policies given by each task as $\{\pi_1, \pi_2, \cdots, \pi_N\}$. The estimated return by the OPE method for $\pi_i$ is denoted as $\hat{\eta}(\pi_i)$. The true return of $\pi_i$ is denoted as $\eta(\pi_i)$. We list the definition of these metrics here.

**Raw Absolute Error** is the average absolute error between the estimated returns and the true returns of $\{\pi_1, \pi_2, \cdots, \pi_N\}$:

$$\text{RawAbsErr} = \frac{1}{N} \sum_{i=1}^{N} |\eta(\pi_i) - \hat{\eta}(\pi_i)|. \tag{42}$$

**Normalized Absolute Error** is the average absolute error after max-min normalization between the estimated returns and the true returns of $\{\pi_1, \pi_2, \cdots, \pi_N\}$:

$$\text{RawAbsErr} = \frac{1}{N} \sum_{i=1}^{N} \left| \frac{\eta(\pi_i) - \min_j \hat{\eta}(\pi_j)}{\max_j \hat{\eta}(\pi_j) - \min_j \hat{\eta}(\pi_j)} - \frac{\hat{\eta}(\pi_i) - \min_j \hat{\eta}(\pi_j)}{\max_j \hat{\eta}(\pi_j) - \min_j \hat{\eta}(\pi_j)} \right|. \tag{43}$$

Table 7: Raw absolute error of each OPE method on DOPE benchmark, averaged over 3 seeds.

| Task Name | Model-free | | | | | Model-based | | | | |
|---|---|---|---|---|---|---|---|---|---|---|
| | FQE | DR | IS | DICE | VPM | EDM | RDM | P-Dreamer | ADM-v1 | ADM-v2 (ours) |
| hopper-exp | 282 | 426 | 106 | 259 | 442 | 295 | 203 | 210 | 89 | **53 ± 7** |
| hopper-med-exp | 252 | 234 | 360 | 266 | 425 | 313 | 101 | 237 | 74 | **37 ± 6** |
| hopper-med | 283 | 307 | 405 | **215** | 433 | 303 | 322 | 294 | 292 | 295 ± 3 |
| hopper-med-rep | 295 | 298 | 438 | 398 | 444 | 76 | 107 | 319 | 90 | **40 ± 9** |
| hopper-rand | 261 | 289 | 412 | **122** | 438 | 324 | 300 | 393 | 276 | 272 ± 11 |
| halfcheetah-exp | 1031 | 1025 | 1404 | 944 | 945 | 1087 | 1049 | **821** | 1011 | 915 ± 64 |
| halfcheetah-med-exp | 1014 | 1015 | 1400 | 1078 | 1427 | 1184 | 961 | 872 | 852 | **711 ± 108** |
| halfcheetah-med | 1211 | 1222 | 1217 | 1382 | 1374 | 969 | 891 | 1343 | 874 | **778 ± 54** |
| halfcheetah-med-rep | 1003 | 1001 | 1409 | 1440 | 1384 | 1009 | 1063 | 1442 | 1020 | **873 ± 73** |
| halfcheetah-rand | 938 | 949 | 1405 | 1446 | 1411 | 1001 | **809** | 1527 | 955 | 856 ± 25 |
| walker2d-exp | 453 | 519 | 405 | 437 | 367 | 458 | 337 | 398 | 250 | **228 ± 32** |
| walker2d-med-exp | 233 | 217 | 436 | 322 | 425 | 446 | 344 | 384 | 242 | **189 ± 20** |
| walker2d-med | 350 | 368 | 428 | 273 | 426 | 393 | 386 | 367 | 213 | **204 ± 46** |
| walker2d-med-rep | 313 | 296 | 427 | 374 | 424 | 358 | 379 | 372 | 194 | **192 ± 22** |
| walker2d-rand | 354 | **347** | 430 | 419 | 440 | 466 | 365 | 407 | 350 | 368 ± 20 |
| Average | 552 | 568 | 712 | 625 | 720 | 549 | 508 | 626 | 452 | **401 ± 33** |

Table 8: Normalized absolute error of each OPE method on DOPE benchmark, averaged over 3 seeds.

| Task Name | Model-free | | | | | Model-based | | | | |
|---|---|---|---|---|---|---|---|---|---|---|
| | FQE | DR | IS | DICE | VPM | EDM | RDM | P-Dreamer | ADM-v1 | ADM-v2 (ours) |
| hopper-exp | 0.55 | 0.83 | 0.21 | 0.51 | 0.86 | 0.58 | 0.40 | 0.41 | 0.18 | **0.10 ± 0.01** |
| hopper-med-exp | 0.49 | 0.46 | 0.70 | 0.52 | 0.83 | 0.61 | 0.20 | 0.46 | 0.15 | **0.07 ± 0.01** |
| hopper-med | 0.55 | 0.60 | 0.79 | **0.42** | 0.85 | 0.59 | 0.63 | 0.57 | 0.57 | 0.58 ± 0.01 |
| hopper-med-rep | 0.58 | 0.58 | 0.86 | 0.78 | 0.87 | 0.15 | 0.21 | 0.62 | 0.17 | **0.08 ± 0.02** |
| hopper-rand | 0.51 | 0.56 | 0.80 | **0.24** | 0.86 | 0.63 | 0.59 | 0.77 | 0.54 | 0.53 ± 0.02 |
| halfcheetah-exp | 0.49 | 0.49 | 0.67 | 0.45 | 0.45 | 0.52 | 0.50 | **0.39** | 0.48 | 0.44 ± 0.03 |
| halfcheetah-med-exp | 0.49 | 0.49 | 0.67 | 0.52 | 0.68 | 0.57 | 0.46 | 0.42 | 0.41 | **0.34 ± 0.05** |
| halfcheetah-med | 0.58 | 0.59 | 0.58 | 0.66 | 0.66 | 0.46 | 0.43 | 0.64 | 0.42 | **0.37 ± 0.03** |
| halfcheetah-med-rep | 0.48 | 0.48 | 0.67 | 0.69 | 0.66 | 0.48 | 0.51 | 0.69 | 0.49 | **0.42 ± 0.04** |
| halfcheetah-rand | 0.45 | 0.45 | 0.67 | 0.69 | 0.68 | 0.48 | **0.39** | 0.73 | 0.46 | 0.41 ± 0.01 |
| walker2d-exp | 0.86 | 0.99 | 0.77 | 0.83 | 0.70 | 0.87 | 0.64 | 0.76 | 0.48 | **0.43 ± 0.06** |
| walker2d-med-exp | 0.44 | 0.41 | 0.83 | 0.61 | 0.81 | 0.85 | 0.65 | 0.73 | 0.46 | **0.36 ± 0.04** |
| walker2d-med | 0.66 | 0.70 | 0.81 | 0.52 | 0.81 | 0.75 | 0.73 | 0.70 | 0.40 | **0.39 ± 0.09** |
| walker2d-med-rep | 0.59 | 0.56 | 0.81 | 0.71 | 0.80 | 0.68 | 0.72 | 0.71 | **0.37** | 0.37 ± 0.04 |
| walker2d-rand | 0.67 | **0.66** | 0.82 | 0.80 | 0.84 | 0.88 | 0.69 | 0.77 | 0.67 | 0.70 ± 0.04 |
| Average | 0.56 | 0.58 | 0.70 | 0.60 | 0.75 | 0.61 | 0.52 | 0.62 | 0.42 | **0.37 ± 0.03** |

Table 9: Rank correlation of each OPE method on DOPE benchmark, averaged over 3 seeds.

| Task Name | Model-free | | | | | Model-based | | | | |
|---|---|---|---|---|---|---|---|---|---|---|
| | FQE | DR | IS | DICE | VPM | EDM | RDM | P-Dreamer | ADM-v1 | ADM-v2 (ours) |
| hopper-exp | -0.33 | -0.41 | 0.37 | -0.08 | 0.21 | 0.46 | 0.67 | -0.26 | 0.83 | **0.90 ± 0.03** |
| hopper-med-exp | 0.01 | -0.08 | 0.35 | 0.08 | 0.17 | 0.54 | 0.83 | -0.25 | 0.90 | **0.95 ± 0.01** |
| hopper-med | -0.29 | -0.31 | -0.55 | 0.19 | 0.13 | 0.58 | 0.77 | 0.51 | 0.88 | **0.89 ± 0.04** |
| hopper-med-rep | 0.45 | 0.05 | -0.16 | 0.27 | -0.16 | 0.91 | 0.93 | -0.23 | 0.93 | **0.97 ± 0.01** |
| hopper-rand | -0.11 | -0.19 | 0.23 | -0.13 | -0.46 | 0.35 | 0.39 | -0.20 | 0.40 | **0.51 ± 0.10** |
| halfcheetah-exp | 0.78 | 0.77 | 0.01 | -0.44 | 0.18 | 0.51 | 0.53 | 0.71 | 0.56 | **0.87 ± 0.10** |
| halfcheetah-med-exp | 0.62 | 0.62 | -0.06 | -0.08 | -0.47 | 0.21 | 0.87 | 0.65 | 0.88 | **0.94 ± 0.02** |
| halfcheetah-med | 0.34 | 0.32 | 0.80 | -0.26 | -0.35 | 0.37 | 0.86 | 0.87 | 0.92 | **0.94 ± 0.01** |
| halfcheetah-med-rep | 0.26 | 0.32 | 0.59 | -0.15 | -0.07 | 0.71 | 0.86 | 0.71 | 0.87 | **0.92 ± 0.02** |
| halfcheetah-rand | -0.11 | -0.02 | -0.24 | -0.70 | 0.27 | 0.75 | **0.81** | -0.58 | 0.63 | 0.78 ± 0.06 |
| walker2d-exp | **0.35** | 0.26 | 0.22 | -0.37 | 0.17 | -0.05 | -0.09 | -0.51 | 0.09 | 0.15 ± 0.11 |
| walker2d-med-exp | 0.25 | 0.19 | 0.24 | -0.34 | 0.49 | 0.21 | 0.27 | -0.88 | 0.61 | **0.70 ± 0.10** |
| walker2d-med | -0.09 | 0.02 | -0.25 | 0.12 | 0.44 | 0.37 | 0.03 | -0.69 | 0.56 | **0.68 ± 0.05** |
| walker2d-med-rep | -0.19 | -0.37 | 0.65 | 0.55 | -0.52 | 0.14 | 0.23 | -0.45 | 0.65 | **0.75 ± 0.03** |
| walker2d-rand | 0.21 | 0.16 | -0.05 | -0.19 | -0.42 | -0.04 | 0.44 | -0.21 | 0.44 | **0.54 ± 0.11** |
| Average | 0.14 | 0.09 | 0.14 | -0.10 | -0.03 | 0.40 | 0.56 | -0.05 | 0.67 | **0.77 ± 0.05** |

**Rank correlation** measures the correlation between the ordinal rankings of the estimated returns and the true returns:

$$\text{RankCorr} = \frac{\text{Cov}([\eta(\pi_i)]_{i=1}^{N}, [\hat{\eta}(\pi_i)]_{i=1}^{N})}{\sigma([\eta(\pi_i)]_{i=1}^{N})\sigma([\hat{\eta}(\pi_i)]_{i=1}^{N})}, \tag{44}$$

Table 10: Regret@1 of each OPE method on DOPE benchmark, averaged over 3 seeds.

| Task Name | Model-free | | | | | Model-based | | | | |
|---|---|---|---|---|---|---|---|---|---|---|
| | FQE | DR | IS | DICE | VPM | EDM | RDM | P-Dreamer | ADM-v1 | ADM-v2 (ours) |
| hopper-exp | 0.41 | 0.34 | 0.06 | 0.20 | 0.13 | 0.07 | 0.10 | 0.49 | 0.09 | **0.05 ± 0.02** |
| hopper-med-exp | 0.42 | 0.34 | 0.10 | 0.16 | 0.12 | 0.14 | 0.07 | 0.58 | **0.06** | 0.06 ± 0.00 |
| hopper-med | 0.32 | 0.32 | 0.38 | 0.18 | 0.10 | 0.42 | 0.16 | 0.27 | **0.06** | 0.08 ± 0.02 |
| hopper-med-rep | 0.18 | 0.34 | 0.88 | 0.16 | 0.21 | 0.16 | 0.04 | 0.28 | 0.06 | **0.03 ± 0.04** |
| hopper-rand | 0.36 | 0.41 | **0.05** | 0.30 | 0.26 | 0.40 | 0.52 | 0.57 | 0.51 | 0.49 ± 0.20 |
| halfcheetah-exp | 0.12 | **0.11** | 0.15 | 0.32 | 0.14 | 0.30 | 0.25 | 0.35 | 0.23 | 0.18 ± 0.15 |
| halfcheetah-med-exp | 0.14 | 0.14 | 0.73 | 0.38 | 0.80 | 0.36 | 0.10 | 0.48 | 0.11 | **0.02 ± 0.04** |
| halfcheetah-med | 0.38 | 0.37 | **0.05** | 0.82 | 0.33 | 0.17 | 0.17 | 0.19 | 0.16 | 0.13 ± 0.09 |
| halfcheetah-med-rep | 0.36 | 0.33 | 0.13 | 0.30 | 0.25 | 0.23 | 0.18 | 0.16 | 0.14 | **0.06 ± 0.13** |
| halfcheetah-rand | 0.37 | 0.31 | 0.31 | 0.81 | **0.12** | 0.41 | 0.20 | 1.00 | 0.32 | 0.24 ± 0.02 |
| walker2d-exp | **0.06** | **0.06** | 0.43 | 0.35 | 0.09 | 0.42 | 0.49 | 0.90 | 0.56 | 0.44 ± 0.32 |
| walker2d-med-exp | 0.22 | 0.30 | 0.13 | 0.78 | 0.24 | 0.43 | 0.10 | 0.80 | 0.04 | **0.02 ± 0.02** |
| walker2d-med | 0.31 | 0.25 | 0.70 | 0.27 | 0.08 | 0.35 | 0.36 | 0.80 | 0.13 | **0.04 ± 0.01** |
| walker2d-med-rep | 0.24 | 0.68 | **0.02** | 0.18 | 0.46 | 0.21 | 0.12 | 0.77 | 0.08 | 0.04 ± 0.00 |
| walker2d-rand | 0.15 | 0.15 | 0.74 | 0.39 | 0.88 | 0.57 | 0.12 | 0.50 | **0.03** | 0.03 ± 0.02 |
| Average | 0.27 | 0.30 | 0.32 | 0.37 | 0.28 | 0.31 | 0.20 | 0.54 | 0.17 | **0.13 ± 0.07** |

where $\mathrm{Cov}(\cdot, \cdot)$ is the covariance between two sequences, and $\sigma(\cdot)$ is the standard deviation of a sequence.

**Regret@k** is the difference between the return of the best policy in the entire set and the return of the best policy in the top-k set (where the top-k set is chosen by estimated returns):

$$\mathrm{Regret}@k = \max_{i \in 1:N} \eta(\pi_i) - \max_{j \in \mathrm{topk}(1:N)} \eta(\pi_i), \tag{45}$$

where $\mathrm{topk}(1:N)$ denotes the indices of the top K policies as measured by estimated returns $\hat{\eta}(\pi)$. We use regret@1 in the experiment.

We report the raw absolute error, normalized absolute error, rank correlation, and regret@1 of each OPE method on the DOPE benchmark in Table 7, Table 8, Table 9, and Table 10, respectively. We select 15 datasets from Hopper, HalfCheetah, and Walker3d, each with five dataset types: expert (exp), medium-expert (exp), medium (med), medium-replay (med-rep), and random (rand). The results of FQE (Le et al., 2019), DR (Jiang & Li, 2016), IS (Kostrikov & Nachum, 2020), DICE (Yang et al., 2020), and VPM (Wen et al., 2020) come from the DOPE (Fu et al., 2021) benchmark.

### G.5 OFFLINE POLICY LEARNING WITH OFF-POLICY EVALUATION

We plot the true return curves obtained from the real environment and the estimated return curves provided by dynamics models during offline policy learning, as shown by Figure 15. Our results reveal that only ADM-v2 produces estimated return curves that closely align with the true returns. Since the real environment is not accessible during offline policy optimization, the accuracy of model-estimated returns becomes crucial. Reliable return estimates are essential both for monitoring policy convergence and for selecting the final policy to be deployed in the real world. As shown in the figure, ADM-v2 satisfies this requirement, highlighting its significance for advancing the practical application of reinforcement learning in real-world scenarios.

### G.6 STUDY ON ROLL-OUT HORIZON AND PENALTY COEFFICIENT

We evaluate the policy performance after learning in ADM-v2, ADM, EDM, RDM, and P-Dreamer on the hopper-medium-v2 task under different roll-out horizons and penalty coefficients and plot the heat map in Figure 16. While setting the roll-out horizon to 1, all models produce policies with poor performance. As the roll-out horizon is gradually increased up to the maximum episode length, the policy performance learned within ADM-v2 tends to improve after appropriately tuning the penalty coefficient. This highlights the potential of full-horizon roll-outs in enhancing policy learning. However, this trend is not as evident for other models, since their predictive accuracy deteriorates in full-horizon roll-outs and thus fails to provide reliable support for policy optimization.

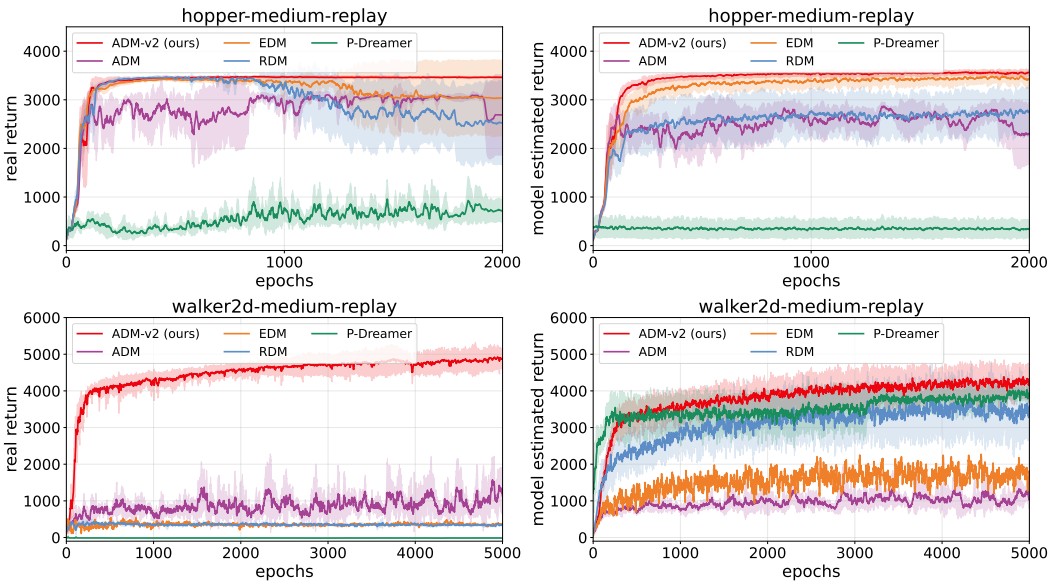

Figure 15: Real return curves vs estimated return curves by dynamics models during offline policy learning on hopper-medium-replay and walker2d-medium-replay tasks, averaged over 5 seeds.

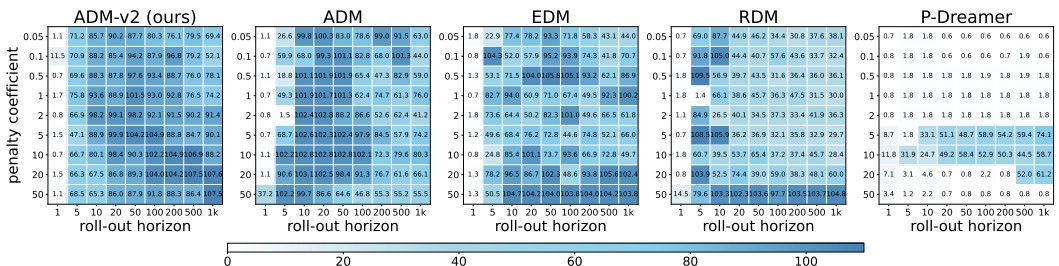

Figure 16: Heat map of policy performance after learning in ADM-v2, ADM, EDM, RDM, and P-Dreamer on `hopper-medium-v2`, under different roll-out horizons and penalty coefficients.

## G.7 ABLATION STUDY ON ENCODER STRUCTURE

Table 11: Normalized scores of ablative settings.

| Task Name | original | 1 layer less | 1 layer more | w/o skip connections | w/o layer normalization |
|---|---|---|---|---|---|
| hopper-medium | $107.6 \pm 0.3$ | $106.1 \pm 0.1$ | $107.2 \pm 0.2$ | $105.1 \pm 0.9$ | $108.0 \pm 0.2$ |
| walker2d-medium | $111.6 \pm 4.8$ | $108.8 \pm 1.3$ | $111.8 \pm 1.1$ | $109.7 \pm 1.4$ | $109.2 \pm 2.5$ |
| hopper-medium-replay | $107.0 \pm 0.5$ | $107.2 \pm 1.1$ | $105.2 \pm 0.8$ | $107.2 \pm 0.3$ | $106.6 \pm 0.2$ |
| walker2d-medium-replay | $106.3 \pm 4.0$ | $111.2 \pm 0.5$ | $107.3 \pm 2.3$ | $109.5 \pm 2.6$ | $106.7 \pm 3.1$ |

The state encoder in ADM-v2 adopts a 5-layer MLP architecture, where the middle three hidden layers are equipped with skip connections (He et al., 2016) and layer normalization (Ba et al., 2016). We aim to investigate the impact of network depth, skip connections, and layer normalization on the algorithm's performance. We design four ablative settings:

- **1 layer less**: Reduce one hidden layer in state encoder.
- **1 layer more**: Add one hidden layer in state encoder.
- **w/o skip connections**: Remove all skip connections in state encoder.
- **w/o layer normalization**: Remove all layer normalization modules in state encoder.

Table 11 presents the results on four tasks: hopper-medium, walker2d-medium, hopper-medium-replay, and walker2d-medium-replay. It can be observed that the architecture of the state encoder has only a minor effect on the algorithm's performance.

## G.8 ROBUSTNESS TO DATA COVERAGE

On the hopper-medium-v2, walker2d-medium-v2, hopper-medium-replay-v2, and walker2d-medium-replay-v2 datasets, we run the algorithms using 50%, 20%, 10%, and 5% of the data, respectively, and compare their performance with that obtained using the full dataset. The results of MOPO, MOPO-fh, ADMPO, ADMPO-fh, and ADM2PO-fh are reported in Table 12.

Table 12: Normalized scores under different data coverage.

| Task Name | MOPO | MOPO-fh | ADMPO | ADMPO-fh | ADM2PO-fh (ours) |
|---|---|---|---|---|---|
| hopper-medium (100% data) | 62.8 | 104.2 | **107.4** | 64.8 | **107.6 ± 0.3** |
| hopper-medium (50% data) | 42.4 | 106.1 | 100.7 | 68.5 | **107.3 ± 0.5** |
| hopper-medium (20% data) | 43.3 | 102.3 | 73.6 | 48.3 | **104.4 ± 0.8** |
| hopper-medium (10% data) | 35.8 | 56.0 | 75.1 | 54.7 | **103.6 ± 0.7** |
| hopper-medium (5% data) | 44.6 | 39.7 | 51.3 | 50.9 | **102.6 ± 1.1** |
| walker2d-medium (100% data) | 84.1 | 27.0 | 93.2 | 58.7 | **111.6 ± 4.8** |
| walker2d-medium (50% data) | 73.5 | 34.1 | 79.5 | 24.2 | **109.5 ± 2.5** |
| walker2d-medium (20% data) | 55.7 | 17.0 | 75.5 | 30.6 | **102.4 ± 1.4** |
| walker2d-medium (10% data) | 11.9 | 10.9 | 61.3 | 22.7 | **99.3 ± 1.7** |
| walker2d-medium (5% data) | 10.6 | 6.6 | 10.6 | -0.5 | **83.9 ± 1.3** |
| hopper-medium-replay (100% data) | 103.5 | 94.2 | 104.4 | 83.4 | **107.0 ± 0.5** |
| hopper-medium-replay (50% data) | 21.7 | 25.7 | 25.9 | 17.1 | **107.4 ± 1.2** |
| hopper-medium-replay (20% data) | 16.0 | 20.6 | 18.1 | 16.4 | **98.7 ± 2.5** |
| hopper-medium-replay (10% data) | 18.2 | 10.9 | 11.9 | 12.5 | **96.6 ± 1.8** |
| hopper-medium-replay (5% data) | 11.3 | 12.3 | 11.7 | 11.5 | **12.7 ± 2.4** |
| walker2d-medium-replay (100% data) | 85.6 | 11.4 | 95.6 | 26.1 | **106.3 ± 4.0** |
| walker2d-medium-replay (50% data) | 76.3 | 10.6 | 67.4 | 23.5 | **98.3 ± 2.7** |
| walker2d-medium-replay (20% data) | 58.7 | 10.2 | 40.3 | 21.6 | **94.9 ± 3.1** |
| walker2d-medium-replay (10% data) | 8.8 | -4.5 | 15.9 | 7.8 | **86.8 ± 2.5** |
| walker2d-medium-replay (5% data) | 3.5 | -3.1 | 16.9 | 7.9 | **36.2 ± 4.6** |

As the amount of available data decreases, the performance of all algorithms tends to decline. Overall, ADM2PO-fh shows the smallest degradation, demonstrating its robustness to data coverage. Notably, on the hopper-medium-v2 and walker2d-medium-v2 datasets, ADM2PO-fh still learns policies with great performance even when trained with only 5% of the data.

## G.9 ROBUSTNESS TO ENVIRONMENTAL PERTURBATIONS

Table 13: Normalized scores under environmental perturbations.

| Task Name | original | state noise $\mathcal{N}(0, 0.001)$ | state noise $\mathcal{N}(0, 0.01)$ | state noise $\mathcal{N}(0, 0.1)$ | smaller gravity (0.9g) | greater gravity (1.1g) |
|---|---|---|---|---|---|---|
| hopper-medium | 107.6 ± 0.3 | 107.0 ± 0.5 | 106.2 ± 1.2 | 29.0 ± 3.1 | 61.2 ± 5.5 | 58.5 ± 3.7 |
| walker2d-medium | 111.6 ± 4.8 | 111.4 ± 4.2 | 111.1 ± 4.6 | 76.7 ± 7.6 | 75.2 ± 6.2 | 68.5 ± 8.4 |
| hopper-medium-replay | 107.0 ± 0.5 | 106.5 ± 0.8 | 105.7 ± 0.6 | 35.9 ± 7.9 | 78.5 ± 4.7 | 81.4 ± 4.5 |
| walker2d-medium-replay | 106.3 ± 4.0 | 105.9 ± 3.4 | 105.5 ± 5.3 | 67.2 ± 6.9 | 91.6 ± 8.7 | 76.9 ± 9.4 |

Table 13 reports the performance of the policies learned by ADM2PO-fh on the four datasets, hopper-medium-v2, walker2d-medium-v2, hopper-medium-replay-v2, and walker2d-medium-replay-v2, under different environment perturbations. We consider five types of perturbations: adding Gaussian noise with a standard deviation of 0.001, 0.01, and 0.1 to the environmental states, decreasing the gravity to 0.9g, and increasing the gravity to 1.1g. As shown in the table, the policy exhibits noticeable performance degradation when the noise level reaches 0.1, while maintaining reasonable robustness under the other perturbations.

Table 14: Normalized scores after offline learning on OGBench tasks.

| Task Name | MOPO | MOBILE | ADMPO | MOPO-fh | RMPO-fh | PDPO-fh | ADMPO-fh | ADM2PO-fh (ours) |
|---|---|---|---|---|---|---|---|---|
| cube-single-play-singletask | 15 | 72 | 65 | 7 | 0 | 0 | 11 | **77 ± 4** |
| cube-double-play-singletask | 5 | 4 | 12 | 0 | 0 | 0 | 0 | **35 ± 5** |
| scene-play-singletask | 8 | 6 | 11 | 0 | 0 | 0 | 3 | **22 ± 6** |
| puzzle-3x3-play-singletask | 16 | 21 | 18 | 0 | 0 | 0 | 0 | **32 ± 5** |
| puzzle-4x4-play-singletask | 0 | 0 | 0 | 0 | 0 | 0 | 0 | **13 ± 8** |

### G.10 OGBENCH RESULTS

Table 14 reports the normalized scores of ADM2PO-fh and several other model-based methods on OGBench (Park et al., 2025) tasks. Most tasks in OGBench are designed for goal-conditioned offline RL algorithms, making them less suitable for the method proposed in this paper. Therefore, we select five single-task manipulation datasets from OGBench to evaluate ADM2PO-fh. As shown, on these challenging long-horizon manipulation tasks, ADM2PO-fh achieves better performance than other offline MBRL algorithms.

### G.11 UNCERTAINTY QUANTIFICATION

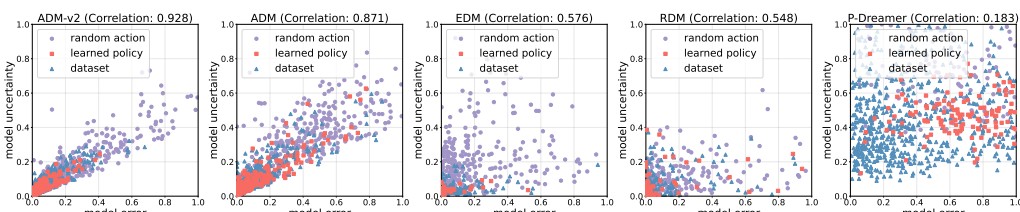

Figure 17: Comparison of uncertainty quantification on `hopper-medium-v2`.

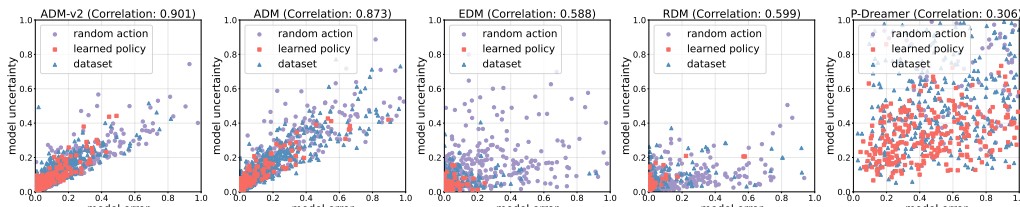

Figure 18: Comparison of uncertainty quantification on `hopper-medium-replay-v2`.

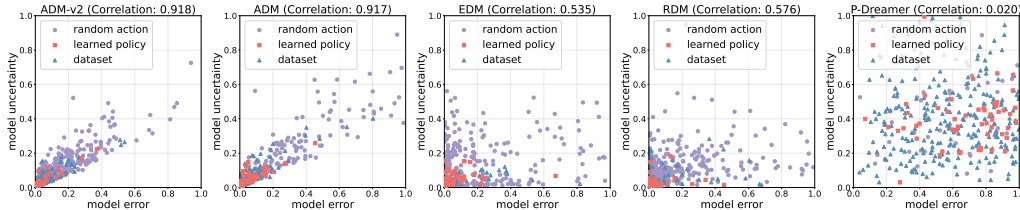

Figure 19: Comparison of uncertainty quantification on `hopper-medium-expert-v2`.

We sample data points in ADM-v2, ADM, EDM, RDM, and P-Dreamer by random action selection, the offline-learned policy, and the dataset behavior. For each sample, we compute the model error and model uncertainty, and scatter the results in Figures 17 to 25. ADM-v2 demonstrates reliable uncertainty quantification, as samples with larger model errors tend to exhibit correspondingly larger model uncertainties. Notably, ADM-v2 achieves a great correlation coefficient between model uncertainty and model error, outperforming all other models. This indicates that $\mathcal{U}^{\mathrm{ADM2}}$ satisfies the assumption of an admissible error estimator (Yu et al., 2020).

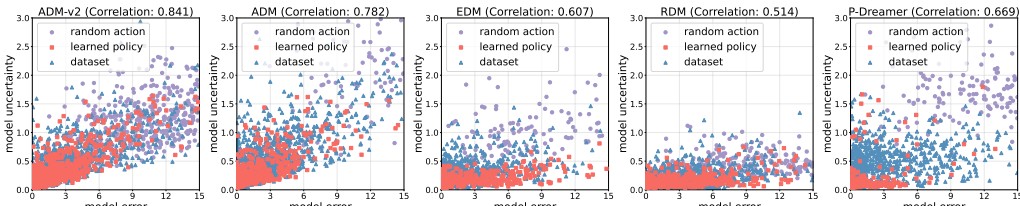

Figure 20: Comparison of uncertainty quantification on `walker2d-medium-v2`.

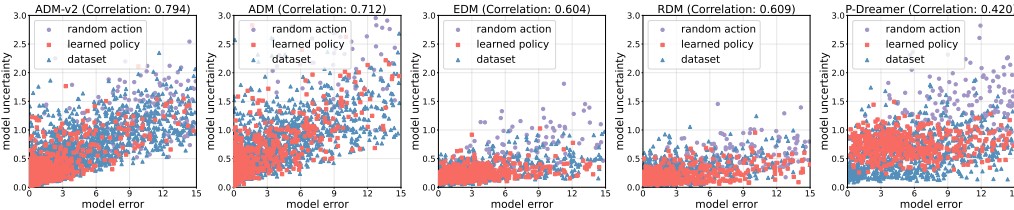

Figure 21: Comparison of uncertainty quantification on `walker2d-medium-replay-v2`.

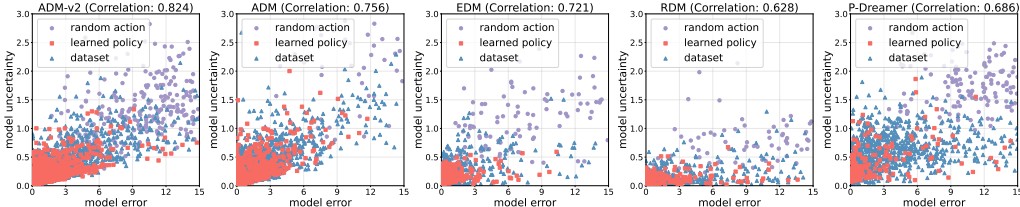

Figure 22: Comparison of uncertainty quantification on `walker2d-medium-expert-v2`.

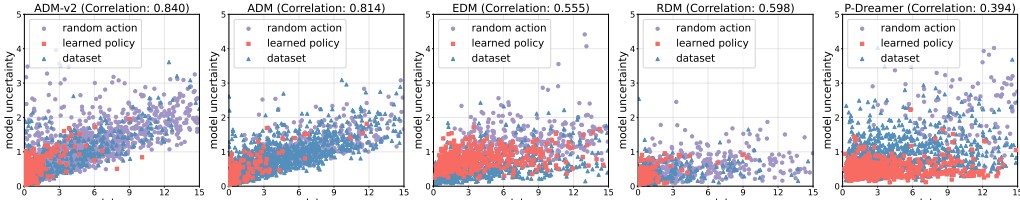

Figure 23: Comparison of uncertainty quantification on `halfcheetah-medium-v2`.

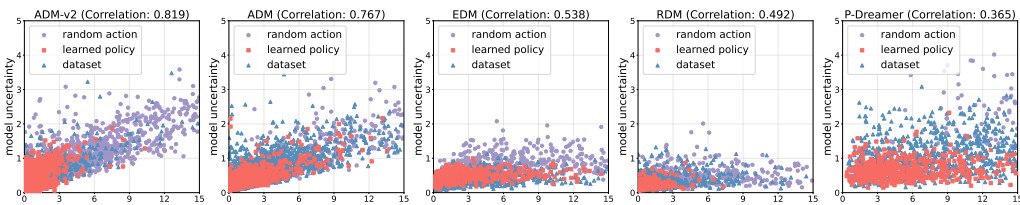

Figure 24: Comparison of uncertainty quantification on `halfcheetah-medium-replay-v2`.

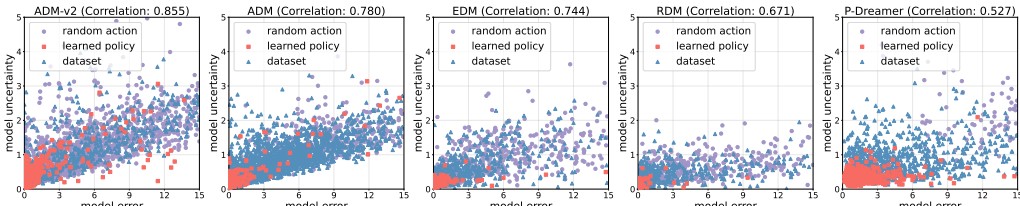

Figure 25: Comparison of uncertainty quantification on `halfcheetah-medium-expert-v2`.

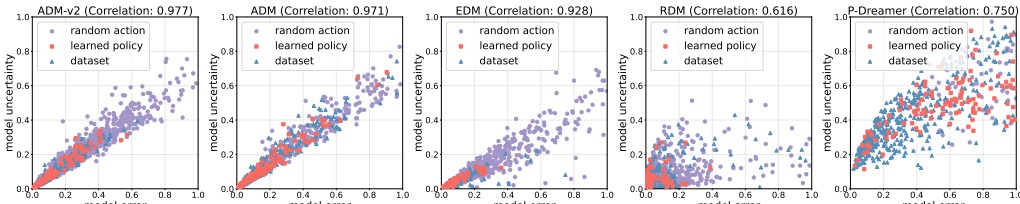

Figure 26: Comparison of uncertainty quantification under short roll-outs (10-step roll-out) on `hopper-medium-v2`.

## G.12 UNCERTAINTY QUANTIFICATION UNDER SHORT ROLL-OUTS

The uncertainty quantification experiments in Appendix G.11 are all conducted under full-horizon roll-outs. Figure 26 presents the uncertainty quantification results on the hopper-medium-v2 task when rolling out only 10 steps. When the roll-out horizon is reduced, the correlation between the uncertainty estimated by these dynamics models and the model error becomes higher than that under long-horizon roll-outs.

## H RELATED WORK

### H.1 OFFLINE MODEL-FREE REINFORCEMENT LEARNING

The interactive exploration inherent in online reinforcement learning is often unsafe and expensive for safety-critical domains, such as robot control (Liu et al., 2024; Yang et al., 2023; Zhang et al., 2026). Offline Model-free Reinforcement Learning (OMFRL) (Fujimoto et al., 2019; Fujimoto & Gu, 2021; Kumar et al., 2020; Zhang et al., 2024c) seeks to learn policies directly from static datasets without access to the environment. To prevent distributional shift from divergence with the behavior policy, OMFRL typically utilizes some form of importance sampling (Precup et al., 2001; Liu et al., 2020), constrains the learned policy through regularization (Kumar et al., 2019; Fujimoto & Gu, 2021), or discourages unreliable actions by adopting pessimistic Q-value optimization methods, such as uncertainty-aware (Agarwal et al., 2020; Zhang et al., 2024a) and conservative estimation (Kumar et al., 2020; Garg et al., 2023). However, these strategies often limit generalization beyond the dataset. By contrast, our model-based approach can extrapolate to unseen state-action pairs, enabling stronger generalization.

### H.2 OFFLINE MODEL-BASED REINFORCEMENT LEARNING

A learned environment model can be used for planning (Camacho & Alba, 2013; Chua et al., 2018; Sun et al., 2026), as well as acting as a simulator for policy learning (Janner et al., 2019; Yu et al., 2020; Lin et al., 2025). Offline Model-based Reinforcement Learning (OMBRL) has shown strong potential in the offline setting, as it can leverage learned dynamics models to expand datasets and improve data efficiency. However, model-generated roll-outs often suffer from distributional shift, which undermines long-horizon predictions. To mitigate this issue, prior works introduce conservative mechanisms and restrict the horizon of roll-outs. MOPO (Yu et al., 2020) and MOReL (Kidambi et al., 2020) penalize rewards with model uncertainty for pessimistic value estimation, while MOBILE (Sun et al., 2023) improves uncertainty quantification via Model-Bellman inconsistency. COMBO (Yu et al., 2021) applies CQL (Kumar et al., 2020) on model-generated samples to constrain out-of-distribution Q-values. RAMBO (Rigter et al., 2022) employs adversarial model learning to enforce conservative estimation, and CBOP (Jeong et al., 2023) integrates adaptive short-horizon rollouts with ensemble variance for robust value estimation. MOREC (Luo et al., 2024) further enhances reliability by designing a reward-consistent dynamics model.

Despite these advances, all methods remain constrained to short or medium horizons due to compounding model errors. In contrast, ADM-v2 introduces a novel architecture that enables roll-outs across the full episode horizon, thereby supporting more effective exploration and policy learning.

### H.3 OFF-POLICY EVALUATION

Off-policy evaluation (OPE) aims to estimate the value of a target policy using only pre-collected data. There are several typical types of OPE methods. Fitted Q-evaluation (FQE) (Le et al., 2019) estimates a value for the Q-function and then utilizes it as the policy value estimator. Importance sampling (IS) (Kostrikov & Nachum, 2020) reweighs observed trajectories by the likelihood ratio between the target and behavior policies. The doubly robust method (Jiang & Li, 2016) combines the importance sampling technique with a value estimator for variance reduction. DICE (Yang et al., 2020) uses a saddle point objective to estimate marginalized importance weights. Variational Power Method (VPM) (Wen et al., 2020) runs a variational power iteration algorithm to estimate the importance weights without the knowledge of the behavior policy. Model-based OPE methods (Hanna et al., 2017; Gottesman et al., 2019; Wang et al., 2024; Chen et al., 2024b;a) learn an approximate dynamics model and evaluate the policy by simulating trajectories within the learned dynamics model.

Model-based methods are appealing for their ability to generalize beyond observed data, but their performance heavily depends on the accuracy of the learned model, which can substantially affect OPE outcomes. Our approach builds on this paradigm and introduces a new model that demonstrates improved effectiveness for OPE.

