# OpenReview forum: "ADM-v2: Pursuing Full-Horizon Roll-out in Dynamics Models for Offline Policy Learning and Evaluation"
_ICLR.cc/2026/Conference — ICLR 2026 Poster_

### Official Review · Reviewer_s9SG · 2025-10-23

**Soundness:** 3
**Presentation:** 3
**Contribution:** 2
**Rating:** 4
**Confidence:** 4

**Summary:**

The paper proposes a modification of the Any-step Dynamics model by Lin et al. Contrary to prior work, the authors propose to remove the per-step conditioning of the hidden state on the initial state in the dynamics model roll-out. They show that this leads to empirically better performance than comparable baselines, and give some theoretical insight into the performance of k-step models.

**Strengths:**

The modification the authors propose is a simple change to the prior work, ADM, removing a conditioning of the dynamics model latent state on the first state in the trajectory. To clarify: I count simple yet impact-full changes as a strength, I think it is a good idea to thoroughly analyze simple and clear design choices.
Experimental evidence supports the benefit of this design decision.
The authors also provide theoretical insight into the benefit of k-step model training in model-based RL that may be of independent interest to the community (see notes below).

**Weaknesses:**

The theoretical contribution is somewhat disjoint from the modification presented here. While it is reasonable to prove that a k-step model can lead to better performance than a one-step model, this proof applies to both the prior work and this work. I think some more targeted analysis of the specific design decision proposed here would have made the paper much stronger. This weakness is currently my main reason for recommending rejection: I do not think that the theoretical contribution serves to justify the design decision proposed here.

Building on this, I overall do not think the paper provides a strong enough intuition for why removing the conditioning on the state helps empirically. I think some more targeted investigation (theoretical or empirical) would greatly strengthen the paper. For example, the analyses presented in Figure 4 do not show ADM-v2 performing significantly better than ADM, yet it leads to farther higher reward. Therefore, investigating e.g. the generalization properties of the method in more detail seem highly relevant to clarify what the modification is doing to the community. Similarly, the analysis in Figure 5 is not tied to performance in the offline RL task, making it hard to assess the impact of an improved correlation.

Some results do not align between prior work and this work. For example, the ADM paper reports a Correlation of 0.98 on the hopper-medium task, while this paper reports 0.871. The score reported in prior work outperforms the performance reported here, and I think it is important for the authors to comment on this. Is this explained by the horizon of the rollout?

**Questions:**

The theoretical performs bound indeed shows a tighter dependency of the error on the roll-out horizon by a factor of $(1-\gamma^m)$. However, this assumes that the m-step error can be controlled to be the same value as the one step error. While this seems to empirically be true in the (very low noise) environments addressed here, I don't see a principled reason for why this should always be the case. Could the authors comment on this? Effectively, is there there a reason to believe that $max_k \delta^k$ does not depend unfavourably on $k$. Basically, is it always reasonable to assume that we can get the k-step error as small as the 1-step error.

While the authors propose full horizon rollouts, they do not test their model as a short horizon model. However, the baseline ADM model performs better as a short horizon than as a long horizon model. Does ADM-v2 actually provide a benefit from doing full horizon rollouts, or would the returns increase even more if it was used at a short horizon? This is a vital experiment that would benefit the narrative of the paper.

---

> ### Author Response · Authors · 2025-11-19
> **Response to Reviewer s9SG**
>
> We are grateful to you for your detailed and constructive feedback.
>
> ---
> **[Q1]**: The paper doesn't provide a strong enough intuition for why removing the conditioning on the state helps empirically. Some more targeted investigation (theoretical or empirical) would greatly strengthen the paper.
> **[A1]**: We present a new experiment in Section 4.6 to provide some insights. In ADM-v2, the hidden state update follows the structure $h_{t+1} = g(h_t, a_t)$, which is consistent with the environment dynamics $s_{t+1} = T(s_t, a_t)$. This structure in ADM-v2 can encourage a correspondence between the transition from $h_t$ to $h_{t+1}$ and the transition from $s_t$ to $s_{t+1}$ during learning. In contrast, the original ADM uses the structure $h_{t+1} = g(s_0, h_t, a_t)$, where $s_0$ is the backtracked state, which does not produce the same effect. In other words, ADM-v2 not only enables multi-step prediction but also preserves the consistency between the transition structure in latent space and the real state transition structure in the environment. Intuitively, under this structure, visualizing the real states and hidden representations should reveal certain correlations. In Section 4.6, we visualize the real states, ADM-v2 hidden representations, and ADM hidden representations using t-SNE. We observe that the spatial distribution of ADM-v2 hidden states closely resembles that of the real states, whereas ADM hidden states do not exhibit this property. This suggests that ADM-v2 has learned a better representation, enabling higher-fidelity long-horizon predictions and thereby improving policy optimization in the model.
>
> ---
> **[Q2]**: Some results do not align between prior work and this work. For example, the ADM paper reports a Correlation of 0.98 on the hopper-medium task, while this paper reports 0.871.
> **[A2]**: In our manuscript, uncertainty quantification experiments are all performed using the full-horizon roll-out setting. In Appendix G.12, we include results for hopper-medium under short roll-out. With a shorter roll-out horizon, the uncertainty estimated by the dynamics models tends to be more trustworthy, leading to a stronger correlation between uncertainty and model error. This underscores the challenge of full-horizon roll-out: only by being able to make accurate long-horizon predictions can one reap the benefits of more thorough exploration of the state space.
>
> ---
> **[Q3]**: The theoretical performs bound indeed shows a tighter dependency of the error on the roll-out horizon. However, this assumes that the m-step error can be controlled to be the same value as the one step error. While this seems to empirically be true in the (very low noise) environments addressed here, I don't see a principled reason for why this should always be the case. Effectively, is there there a reason to believe that $\max_k \delta^k$ does not depend unfavourably on $k$. Basically, is it always reasonable to assume that we can get the k-step error as small as the 1-step error.
> **[A3]**: We do not actually assume that the m-step error can be controlled to be equal to the one-step error. In our derivation, we only use the basic inequalities $\delta^1 < \max_k \delta^k$, $\delta^2 < \max_k \delta^k$, $\ldots$, $\delta^m < \max_k \delta^k$ for scaling. The performance bound we derive is $\frac{2\gamma r_{\mathrm{max}}\delta_{\mathrm{max}}}{(1-\gamma)(1-\gamma^m)}+\frac{2\gamma r_{\mathrm{max}}\epsilon_\pi}{(1-\gamma)^2}+\frac{4r_{\mathrm{max}}\epsilon_\pi}{1-\gamma}$, and by setting $m=1$ we obtain $\frac{2\gamma r_{\mathrm{max}}\delta^1}{(1-\gamma)^2}+\frac{2\gamma r_{\mathrm{max}}\epsilon_\pi}{(1-\gamma)^2}
> +\frac{4r_{\mathrm{max}}\epsilon_\pi}{1-\gamma}$, which corresponds to the bound for a single-step dynamics model. Comparing the two bounds, we find that when condition $\delta_{\mathrm{max}}<\frac{\delta^1(1-\gamma^m)}{1-\gamma}$ holds, the bound for ADM-v2 is tighter. The experiments in Figure 3(a) are designed to illustrate whether this condition can be satisfied. Here, we do not impose any requirement on the trend of $\delta^k$. Whether $\delta^k$ increases with $k$ or not does not affect our derivation, nor does it affect our choice of an $m$ that satisfies the condition.

---

> > ### Author Response · Authors · 2025-11-19
> >
> > **[Q4]**: Does ADM-v2 actually provide a benefit from doing full horizon rollouts, or would the returns increase even more if it was used at a short horizon?
> > **[A4]**: This is a highly meaningful question. In Section 4.5, we plot the performance curves of ADM2PO-fh as the roll-out horizon varies (as well as those of the other four baseline dynamics models) on the hopper-medium and walker2d-medium tasks. The experimental results show that the performance of ADM2PO-fh tends to improve as the roll-out horizon increases, whereas the other dynamics models do not exhibit such a trend. As indicated by our evaluations of the predictive capability of these dynamics models, the long-horizon prediction ability of the other models is poor, and therefore they cannot benefit from increasing the roll-out horizon.

---

> > > ### Comment · Reviewer_s9SG · 2025-11-22
> > > **Quick reply**
> > >
> > > Dear authors,
> > >
> > > thanks for your detailed response. I haven't replied in detail yet or updated my score since I wanted to go through the proofs more carefully again after your comment. I'll hopefully be able to send a reply on Monday. Just wanted to let you know that your comments are not being ignored, I know how frustrating the silence can be from the authors point of view.

---

> > > > ### Author Response · Authors · 2025-11-23
> > > >
> > > > Thank you for the reply! We really appreciate your efforts and look forward to your detailed feedback.

---

> > > > > ### Comment · Reviewer_s9SG · 2025-11-24
> > > > > **Thank you**
> > > > >
> > > > > Dear authors,
> > > > >
> > > > > I believe the additional experiments provided greatly strengthen the paper and are sufficient for me to recommend acceptance.
> > > > >
> > > > > Regarding the theoretical results: You are correct in that I misstated the theoretical results somewhat. There is a gap between the any-step TD error and the 1-step one that allows more error in any n-step prediction compared to the 1-step one.
> > > > >
> > > > > However, I think there is still some weirdness here. The paper makes the case for a full horizon rollout, and as your new experiments show, this is indeed a good idea in the empirical setting considered. However, your theory gives a method for evaluating what $m$-step model to use which is dependent on $\max \delta^k$. I strongly suspect that $\max \delta^k$ will depend unfavourably on the horizon in many environments, and so in these cases a full horizon rollout should be a bad idea.
> > > > >
> > > > > As $m\rightarrow\infty$, the presented bound simplifies to $\delta^1 \frac{1}{1-\gamma}$ if I am not mistaken, so we require the TV at any step to be below this number. In the introduction, one of the purported benefits of a full-horizon rollout is to allow a policy to explore. However, this will now require the model to generalize to potentially very hard ood cases, and fullfil the $\frac{1}{1-\gamma}$ error condition.
> > > > >
> > > > > So my issue with the theory somewhat remains: As presented it doesn't really justify a full horizon rollout, but gives a more concrete guideline for choosing $m$ (albeit depending on a quantity that is normally almost impossible to estimate, TV). As presented, it actually makes me more skeptical that full horizon rollout fidelity can be accomplished in more complicated scenarios, even with the presented method. In addition, you need this condition to hold under any policy, not just the data gathering policy, correct? The paper here has a similar issue to the MBPO paper: while the theory is correct as far as I can tell, it doesn't cleanly connect to the story being told in the rest of the paper.
> > > > >
> > > > > Now, all that being said, I still think that given the overall correctness of both theory and the results presented, I'm happy to recommend acceptance.

---

> > > > > > ### Author Response · Authors · 2025-11-25
> > > > > >
> > > > > > Thank you very much for your feedback. We greatly appreciate your careful review and constructive comments, which have been incredibly helpful to us.
> > > > > >
> > > > > > We agree with your assessment regarding our theoretical analysis. As you pointed out, the connection between the theory presented in the manuscript and the motivation (as well as the final empirical results) is somewhat limited. Indeed, the current theory primarily serves as a guideline for selecting an appropriate $m$.
> > > > > >
> > > > > > We believe that developing better theoretical tools to analyze any-step dynamics models and full-horizon roll-outs would be highly beneficial to the model-based RL community, and this will certainly be a focus of our future efforts.
> > > > > >
> > > > > > Thank you again for the time and effort you dedicated to reviewing our work.

---

### Official Review · Reviewer_ri32 · 2025-10-30

**Soundness:** 3
**Presentation:** 3
**Contribution:** 3
**Rating:** 6
**Confidence:** 3

**Summary:**

This paper proposes ADM-v2 (encoding the initial state (s_0) as a hidden state (h_0), combining any-step prediction with PARoll parallel roll-out, and using any-step uncertainty as a conservative penalty). The approach is direct and feasible in engineering, demonstrating significant advantages in full-horizon generation and evaluation of offline RL. The paper has practical value, but currently lacks several key ablation techniques, details, and more rigorous robustness experiments.

**Strengths:**

1.Focusing on error accumulation and parallelization bottlenecks in long horizon roll-out, which are core pain points in offline MBRL.
2.Encoding (s_0) as (h_0) and removing backtracking, the parallelization approach (PARoll) is easily implemented in engineering.
3.The any-step supervised design significantly improves sample utilization, which is a reasonable path to enhance long-horizon capabilities.
4. Includes roll-out error curves, throughput (samples/sec), offline evaluation (DOPE), and downstream policy performance (D4RL/NeoRL), providing multi-dimensional evidence.
5. Using any-step variance as a conservative penalty is practically meaningful for offline learning security.

**Weaknesses:**

1.The design of compressing (s_0) into (h_0) is highly dependent on encoder selection and capacity. The paper lacks sufficient ablation and quantitative analysis of this (e.g., the impact of different encoder capacities, aux losses, and skip connections).

2.Relies on ensembles, but lacks horizon-wise calibration evaluation (the relationship between ensemble variance and true error should be shown more systematically); the computational/memory overhead and trade-offs of ensembles are not adequately discussed.

3.The paper claims a significant increase in throughput, but lacks reproducible PARoll pseudocode, batch organization examples, and memory consumption analysis; throughput comparisons may also be affected by implementation optimizations (a fairer baseline setting explanation is needed).

4.Most metrics are based on open-loop any-step evaluation, but the final policy will be implemented under closed-loop conditions; the paper should supplement closed-loop rollout results or use hybrid training (scheduled sampling / one-step + any-step) to validate stability.

5.Lacks robustness testing for insufficient data coverage or policy distribution drift (e.g., perturbations to initial states or actions, domain-shift testing).

6. The paper needs to provide a more complete list of hyperparameters (ensemble size, k-sampling strategy, encoder/GRU structure, number of training steps, seed), as well as a commitment to release the code/training script.

**Questions:**

See Weaknesses.

---

> ### Author Response · Authors · 2025-11-19
> **Response to Reviewer ri32**
>
> We appreciate your careful evaluation and helpful suggestions. We address your concerns below.
>
> ---
> **[Q1]**: The design of compressing (s_0) into (h_0) is highly dependent on encoder selection and capacity. The paper lacks sufficient ablation and quantitative analysis of this (e.g., the impact of different encoder capacities, aux losses, and skip connections).
> **[A1]**: Thank you very much for your suggestion. We have added ablation experiments on the encoder architecture in Appendix G.7. We found that increasing or decreasing the number of encoder layers, removing skip connections, or removing layer normalization has only a minor impact on the final policy performance. The specific structure of the encoder appears to have little intrinsic relevance to the effectiveness of our method.
>
> ---
> **[Q2]**: Relies on ensembles, but lacks horizon-wise calibration evaluation (the relationship between ensemble variance and true error should be shown more systematically); the computational/memory overhead and trade-offs of ensembles are not adequately discussed.
> **[A2]**: Sorry for any confusion caused by the phrasing in our manuscript. Our approach does not rely on ensembles—rather, the uncertainty estimation is derived from the diversity produced by any-step predictions.
>
> ---
> **[Q3]**: The paper claims a significant increase in throughput, but lacks reproducible PARoll pseudocode, batch organization examples, and memory consumption analysis; throughput comparisons may also be affected by implementation optimizations (a fairer baseline setting explanation is needed).
> **[A3]**: We provide the pseudocode of PARoll in Appendix D.2, which is also applicable to roll-outs with a batch size dimension. The pseudocode for roll-outs of other dynamics models is presented individually in Appendix E. It can be seen that the roll-out process of all dynamics models follows the same framework, consisting of three steps: action sampling by the policy, forward computation in the dynamics model along with uncertainty estimation, and then repeating for H steps. When measuring roll-out throughput, we control the roll-out batch size and horizon to be consistent across all dynamics models to ensure fairness, as explained in the text. The memory consumption analysis is provided in Appendix G.3, showing that ADM-v2 does not require more GPU memory.
>
> ---
> **[Q4]**: Most metrics are based on open-loop any-step evaluation, but the final policy will be implemented under closed-loop conditions; the paper should supplement closed-loop rollout results or use hybrid training (scheduled sampling / one-step + any-step) to validate stability.
> **[A4]**: We provide the training pseudocode for ADM-v2 in Appendix D.1, where the any-step prediction loss is taken into account during training. That is, for any $k \in [1, m]$, the corresponding k-step loss is considered. All ADM-v2 training in the experiments reported in our manuscript follows this procedure. Here, we provide a clarification regarding the metrics shown in Figure 3(a). During training, we consider prediction losses from 1-step to 10-step simultaneously, and the reported error is computed based on the 1-step to 10-step predictions.
>
> ---
> **[Q5]**: Lacks robustness testing for insufficient data coverage or policy distribution drift (e.g., perturbations to initial states or actions, domain-shift testing).
> **[A5]**: Thank you very much for your suggestion. In Appendix G.8, we show the performance degradation when the dataset size is reduced to 50%, 20%, 10%, and 5%. It can be seen that ADM2PO-fh is highly robust to limited data coverage. In Appendix G.9, we present the performance degradation of ADM2PO-fh under various environmental perturbations. As expected, performance inevitably drops when the environment is noisy or its parameters change, which is a major challenge in RL. Without special treatment, the policy struggles to adapt to environmental changes, and adaptive RL is beyond the scope of this paper.
>
> ---
> **[Q6]**: The paper needs to provide a more complete list of hyperparameters (ensemble size, k-sampling strategy, encoder/GRU structure, number of training steps, seed), as well as a commitment to release the code/training script.
> **[A6]**: We describe the detailed structures of the encoder and GRU in Appendix D.5, and provide all algorithm-related hyperparameter settings in Appendix F.1.

---

### Official Review · Reviewer_BzbB · 2025-11-01

**Soundness:** 3
**Presentation:** 4
**Contribution:** 3
**Rating:** 8
**Confidence:** 4

**Summary:**

This work creates an extension of the Any-step Dynamics Model (ADM, ICLR2025). This is a model-based RL method that creates a dynamics model that can directly predict m steps forward by feeding in a sequence of the next m actions (there existed some previous works that did this, but the ADM paper made it more practical). The ADM model has a GRU recurrent network core that has a transition decoder to predict the states from the GRU internal state. ADMv2 modifies this by adding a State encoder before the GRU, so that the state can be embedded beforehand into a better representation. This is trained end-to-end. Another addition is the PARoll prediction mechanism to make multiple forward predictions more practical compared to the method used in the original ADM. To give a bit of background: as ADM allows predicting for any number of steps forward (less than m steps forward), we can construct multiple different predictions for the next state, based on from how many steps backward we will aim to predict the next state. ADM uses these multiple predictions to form an uncertainty estimate of for the next state prediction akin to ensemble methods. However, the original ADM performed this ensemble-like prediction in an inefficient manner, so the current work developed a PARoll framework, that essentially keeps m separate prediction streams for the different forward prediction lengths and runs these in parallel.

For offline policy learning, the uncertainty estimate (specifically the variance) is added to the rewards in the q-updates as a penalty with $-\beta$ multiplier. They also give some theoretical performance bound.

Experimentally, the work evaluates on D4RL, NeoRL offline RL benchmarks, and on the DOPE off-policy evaluation benchmark. The method achieves new state of the art results on D4RL and NeoRL and also had the top results on DOPE. They achieve these results while doing full episode rollouts with their model (i.e. 1000 step rollouts, starting from sampled states from the buffer).

In addition, they tested points such as the model prediction accuracy (which improved), rollout throughput (which improved greatly compared to ADM, but was lower than the ensemble-based method), and also checked correlation between model errors and the uncertainty quantification (there was a good correlation for their method).

**Strengths:**

- The empirical performance appears very strong.

- The writing was good.

- Many metrics were tested, e.g., accuracy, uncertainty quantification, multiple benchmarks.

**Weaknesses:**

- Novelty is that not high as it mostly an extension of ADMv1.

**Questions:**

One way to strengthen the paper would be to also evaluate on OGBench:
https://arxiv.org/pdf/2410.20092

Regarding multi-step predictions, there are also other earlier related works:

Learning Awareness Models, Brandon Amos et al. (ICLR2018)
https://arxiv.org/pdf/1804.06318

Learning Latent Dynamics for Planning from Pixels, Hafner et al. (ICML2019)
https://arxiv.org/pdf/1811.04551

On the other hand, you already cited the works of Asadi (2018). But I think it would be better to cite such a work in the introduction, rather than in the appendix.

The original ADM paper also evaluated such correlations, and in their case, the ensemble model correlations are better than the ones reported in this paper. Where did the discrepancy come from?

---

> ### Author Response · Authors · 2025-11-19
> **Response to Reviewer BzbB**
>
> We sincerely thank you for your thoughtful comments and constructive suggestions. We address your concerns below.
>
> ---
> **[Q1]**: Evaluation on OGBench.
> **[A1]**: Appendix G.10 reports the performance comparison between ADM2PO-fh and several offline model-based baselines on OGBench. The benchmark consists of demanding long-horizon robotic manipulation tasks in which edge states become observable only at task completion. Across these settings, ADM2PO-fh achieves better performance than the other model-based approaches.
>
> ---
> **[Q2]**: Issues on some related works.
> **[A2]**: Thank you very much for the reminder. We have added citations to papers [1][2] in the Related Works section of the main text, and we have also incorporated a description of the works of Asadi (2018) in the Introduction.
>
> ---
> **[Q3]**: The original ADM paper also evaluated the correlation between the estimated uncertainty and the model error, and in their case, the ensemble model correlations are better than the ones reported in this paper. Where did the discrepancy come from?
> **[A3]**: In our manuscript, all uncertainty quantification experiments are conducted under the full-horizon roll-out setting. We have added the uncertainty quantification results on hopper-medium under short roll-out in Appendix G.12. When the roll-out horizon is short, the uncertainty estimated by the dynamics models is more reliable, and therefore the correlation between uncertainty and model error is higher. This, in fact, highlights that full-horizon roll-out is a challenging task. Only with the capability of long-horizon prediction can one fully benefit from deeper exploration of the state space.
>
> [1] Learning Awareness Models, Brandon Amos et al. (ICLR2018)
> [2] Learning Latent Dynamics for Planning from Pixels, Hafner et al. (ICML2019)

---

> > ### Comment · Reviewer_BzbB · 2025-11-27
> > **Review is addressed well**
> >
> > Thank you for the response. I appreciate the added uncertainty quantification, added literature, and added experiments on OGBench. The results on OGBench are not as strong compared to the results on the other environments that you consider (you outperform other model-based methods, but there are still stronger methods than yours, e.g., in the original OGBench paper), but nevertheless are informative.
> >
> > I increased the soundness score, but will keep the rest the same, as my rating is already strong.

---

> > > ### Author Response · Authors · 2025-11-27
> > > **Thank you**
> > >
> > > We sincerely appreciate the time and effort you have dedicated to the review process. Your valuable comments and suggestions have helped us significantly improve our manuscript. We are deeply encouraged by your feedback.

---

### Official Review · Reviewer_6xfG · 2025-11-02

**Soundness:** 4
**Presentation:** 4
**Contribution:** 3
**Rating:** 6
**Confidence:** 3

**Summary:**

The authors propose ADM-v2, a novel dynamics model architecture that builds upon the "any-step" direct prediction paradigm of the original ADM. The key innovation is decoupling the initial state from the recurrent processing of the action sequence. ADM-v2 allows full-horizon roll-outs and achieves new state-of-the-art performance on both D4RL and NeoRL benchmarks.

**Strengths:**

[S1] ADM-v2 achieves high empirical performance while maintaining consistent uncertainty estimation and fast dynamics rollout. It also supports full-horizon expansion, which opens up new possibilities for learning long-horizon tasks.

**Weaknesses:**

[W1] The evaluation only consists of MuJoCo locomotion tasks, which are mostly reactive. Model-free and model-based methods with short-horizon prediction also achieves high performance on MuJoCo. It is unknown that if full-horizon rollout is required.

[W2] This work is an incremental (albeit highly effective and important) improvement focused on a refined architecture and a new roll-out algorithm.

**Questions:**

[Q1] How does ADM2PO (short, limited horizon rollout of ADM2) perform compared to the reported ADM2PO-fh (full horizon rollout)?

[Q2] How does ADM2PO perform in tasks that truly require long-horizon rollout and planning?

---

> ### Author Response · Authors · 2025-11-19
> **Response to Reviewer 6xfG**
>
> We would like to express our gratitude to you for your valuable comments and questions. We address your concerns below.
>
> ---
> **[Q1]**: How does ADM2PO (short, limited horizon roll-out of ADM2) perform compared to the reported ADM2PO-fh (full horizon roll-out)?
> **[A1]**: This is a very meaningful question. In Section 4.5, we plot how the performance of ADM2PO-fh varies with the roll-out horizon (along with the curves of the other four baseline dynamics models) on the hopper-medium and walker2d-medium tasks. The results show that the performance of ADM2PO-fh tends to improve as the roll-out horizon increases, whereas the other dynamics models do not exhibit such a trend. As indicated by our evaluation of their predictive capabilities, the long-horizon prediction accuracy of the other dynamics models is relatively poor, preventing them from benefiting from longer roll-out horizons.
>
> ---
> **[Q2]**: How does ADM2PO perform in tasks that truly require long-horizon roll-out and planning?
> **[A2]**: We present a comparison of ADM2PO-fh against other offline model-based algorithms on OGBench in Appendix G.10. The evaluated tasks are challenging long-horizon robotic manipulation tasks, where edge states are only accessible upon task completion. ADM2PO-fh demonstrates consistently better performance than the other model-based methods on these tasks.
>
> ---
> **[Q3]**: Is full-horizon roll-out required?
> **[A3]**: We believe that full-horizon roll-out serves two major purposes. First, for both locomotion and manipulation tasks, a long horizon enables the policy to explore complete trajectories, especially the edge states, which are crucial for accurate Q estimation. Without feedback from these edge states to close the bootstrap loop, the Q-values would otherwise suffer from bias. Second, when the dynamics model is used for off-policy evaluation, full-horizon roll-out facilitates a more complete assessment of the policy.

---

### Author Response · Authors · 2025-11-19
**General Response**

We sincerely thank all reviewers for their thoughtful comments and constructive suggestions. Their detailed feedback has significantly helped us improve the manuscript. In response to the major concerns raised by all reviewers, we have added some content to the manuscript. The new content is highlighted in blue for easy identification. The additions include:

- We present the normalized scores under different roll-out horizons in Section 4.5 (6xfG, s9SG)
- We present a visualization experiment of real states and hidden representations in Section 4.6 to explain why ADM-v2 performs better (s9SG)
- We present the detailed structure of the encoder in Appendix D.5 (ri32)
- We provide roll-out pseudo-codes for each baseline dynamics model in Appendix E (ri32)
- We list all hyper-parameters settings of ADM-v2 in Appendix F.1 (ri32)
- We present the memory consumption analysis during roll-out in Appendix G.3 (ri32)
- We add an ablation study on encoder structure in Appendix G.7 (ri32)
- We present the performance under limited data coverage in Appendix G.8 (ri32)
- We present the performance under different environmental perturbations in Appendix G.9 (ri32)
- We present the OGBench results to show the performance of ADM-v2 in long-horizon manipulation tasks in Appendix G.10 (6xfG, BzbB)
- We present the uncertainty quantification results under short roll-outs in Appendix G.12 (BzbB, s9SG)

We also provide point-by-point responses to the reviewers’ concerns following each reviewer’s comments.

---

> ### Author Response · Authors · 2025-12-01
> **Follow-up Summary for the Area Chair**
>
> Given the unexpected interruption of the rebuttal phase due to the information leakage incident, we provide this summary to assist the AC in reviewing how we have addressed the reviewers' concerns.
>
> **Crucially, before the discussion was halted:**
> * **Reviewer s9SG** explicitly confirmed that his concerns were addressed and expressed willingness to recommend **acceptance**.
> * **Reviewer BzbB** also responded, confirming that our revisions had resolved his concerns.
>
> Below is a summary of the main concerns from each reviewer and our corresponding responses.
>
> ---
>
> ### Reviewer 6xfG
> **Concerns:**
> * Comparison between short-horizon (ADM2PO) and full-horizon (ADM2PO-fh) performance.
> * Performance on tasks truly requiring long-horizon roll-out.
>
> **Our Response:**
> * **Horizon Analysis:** We added analysis in **Section 4.5** showing that our method's performance tends to improve as the roll-out horizon increases.
> * **Long-Horizon Benchmarks:** We evaluated performance on OGBench (**Appendix G.10**), which specifically targets long-horizon manipulation.
>
> ---
>
> ### Reviewer BzbB
> **Concerns:**
> * Required evaluation on OGBench benchmarks.
> * Missing citations for specific related works (Amos et al., Hafner et al.).
> * Noted discrepancies in uncertainty correlation results compared to prior work.
>
> **Our Response:**
> * **Added OGBench Evaluation:** We added **Appendix G.10**, comparing our method against offline model-based baselines on demanding OGBench tasks.
> * **Updated Citations:** We incorporated the suggested references in the Introduction and Related Works sections.
> * **Clarification on Uncertainty Results:** We clarified that the discrepancy comes from different roll-out horizon settings. We added **Appendix G.12** to show the results under short roll-out, largely consistent with the original ADM paper.
>
> ---
>
> ### Reviewer ri32
> **Concerns:**
> * Lack of ablation studies on encoder architecture.
> * Requested reproducible pseudo-codes, hyper-parameter settings, memory analysis, and robustness testing (data coverage/drift).
>
> **Our Response:**
> * **Ablation Studies:** We added **Appendix G.7**, demonstrating that performance is robust to changes in the encoder structure.
> * **Pseudo-codes:** We added more detailed pseudo-codes (**Appendix D.2, E**)
> * **Hyper-parameters:** We replenish hyper-parameters settings of ADM-v2 (**Appendix F.1**)
> * **Memory Analysis:** We added a memory consumption analysis (**Appendix G.3**), confirming no extra GPU memory overhead.
> * **Robustness Experiments:** We added **Appendix G.8** (data reduction down to 5%) and **Appendix G.9** (environmental perturbations), showing the robustness of the method.
>
> ---
>
> ### Reviewer s9SG
> **Concerns:**
> * Requested stronger intuition/empirical investigation on why removing state conditioning ($s_0$) helps.
> * Noted discrepancies in uncertainty correlation results compared to prior work.
> * Questions on theoretical performance bounds.
> * Questions on the performance under short roll-out.
>
> **Our Response:**
> * **New Experiments:** We added **Section 4.6**, using t-SNE to visualize latent spaces. This demonstrates that ADM-v2 maintains a structure consistent with real environmental dynamics, whereas the original ADM does not.
> * **Clarification on Uncertainty Results:** We clarified that the discrepancy comes from different roll-out horizon settings. We added **Appendix G.12** to show the results under short roll-out, largely consistent with the original ADM paper.
> * **Theoretical Clarification:** We clarified the derivation of the performance bound.
> * **Horizon Analysis:** We added analysis in **Section 4.5** showing that our method's performance tends to improve as the roll-out horizon increases.
>
> ---
>
> ### Remark
> We have addressed all reviewer comments by adding comprehensive experiments (OGBench, robustness, ablation) and clarifications. Two reviewers (**s9SG**, **BzbB**) explicitly acknowledged these improvements before the information leakage incident. We believe the revised manuscript now provides comprehensive evidence, resolves all reviewer concerns, and reflects the contribution of our work. We sincerely appreciate your tremendous efforts and valuable time in handling our submission.

---

### Meta-Review · Area_Chair_dCMt · 2026-01-05

**Summary:**

This paper proposes ADM-v2, which extends the previous work Any-step Dynamics Model (ADM) by decoupling the dynamics model hidden state update from the initial state and introducing PARoll to speed up any-step rollout with uncertainty estimation.

Strengths:
- Strong empirical performance
- Comprehensive experimental evaluations

Weaknesses:
- Novelty is limited
- Weak connection between the theory and the design choice

**Reviewer Concerns:**

The authors' rebuttal provided clarifications and additional experimental details/results that effectively addressed most of the raised concerns, including the experiments beyond MuJoCo tasks from reviewer 6xfG, more experimental details and ablation studies from reviewer ri32, and clarifications of the theoretical results, motivation, and mismatch with prior work from reviewer s9SG.

While some reviewers raised concerns about the technical novelty, noting that the modification over ADM is relatively simple, there was consensus that ADM-v2 is an important extension with substantial practical value.

**Reviewer Scores:**

Reviewers 6xfG, BzbB may maintain their score

Reviewer ri32 may increase the score to 8 as most concerns are about more ablation studies and clarifications, which I think have been addressed in the rebuttal

Reviewer s95G may increase the score to 6 as all raised concerns except the weak theory-design connect have been addressed, which was also indicated by the reviewer

---

### Decision · Program_Chairs · 2026-01-26

Accept (Poster)